# Research on the improvement path of grassroots social governance innovation performance in China——Qualitative comparative analysis based on 35 cases

Nana Song[1‡], Longshun Xu[2‡]*, Xiansheng Chen[1]*, Huange Xu[3], Shuoliang Jiang[1]

1 School of Public Economics and Management, Shanghai University of Finance and Economics, Shanghai, China, 2 School of Public Administration and Society, Jiangsu Normal University, Xuzhou, Jiangsu, China, 3 School of Public Administration, Southwest Jiaotong University, Chengdu, China

‡ NS and LX are co-first authors to this work.
* chenxs2002@163.com (XC); xlsh19910107@163.com (LX)

**Data Availability Statement:** All relevant data are within the paper and its Supporting Information files.

## Abstract

With the rapid development of China's economy and society, the innovation of grassroots social governance has become increasingly important. This paper constructs 35 grassroots social governance innovation samples. Using the TOE theoretical framework and a fuzzy set qualitative comparative analysis (fsQCA), this paper analyzes the joint effects and interactive relationships of multiple factors on grassroots social governance innovation performance from three dimensions: technology, organization, and environment. The research reveals that internal environmental openness is a necessary condition for achieving high innovation performance in grassroots social governance, and proposes four grouping models that affect the performance of grassroots social governance innovation. This paper explores the inner logic of grassroots social governance innovation from a histological perspective, and on this basis proposes an adaptive path to enhance the performance of grassroots social governance innovation.

## 1 Introduction

In recent years, Grassroots social governance has attracted wide attention [1]. Grassroots society refers to the lowest and fundamental level of social organization and activities. It typically refers to the basic social units composed of residents, communities, villages, streets, etc., in urban or rural areas [2]. Grassroots society is characterized by direct engagement with residents, serving them, and addressing their practical issues [3]. China has made significant efforts to promote grassroots social governance innovation in order to modernize its social governance system and enhance its governance capacity. This has been further advanced by the establishment of awards such as the "China Local Government Innovation Award" and the "China Top Ten Social Governance Innovation Awards". Although governments at all levels have promoted the practice of grassroots social governance innovation, due to the imperfect

**Funding:** Youth Project of Humanities and Social Sciences Foundation of Ministry of Education: Research on the Mechanism and Policy Effect of Cross-basin Horizontal Ecological Compensation on Enterprises' High-quality Development (grant number 22YJC630172) Partial financial support was received from the Shanghai Philosophy and Social Science Planning Fund Project: "Pudong New Area to build a Leading area of Socialist Modernization Construction" (grant number 2021BZZ002). The funders had no role in study design, data collection and analysis, decision to publish, or preparation of the manuscript.

**Competing interests:** The authors have declared that no competing interests exist.

system and mechanism, imperfect laws and regulations, and blocked participation channels, some grassroots social governance innovation projects have a short-lived phenomenon, and it is difficult to ensure the efficiency and sustainability of innovation projects [4]. Therefore, the goal that this article addresses is: why do grassroots social governance innovation projects suffer from the same fate of being ineffective despite taking different paths? How to improve the practical efficiency of grassroots social governance innovation projects?

The academic circles mainly study the innovation of grassroots social governance from three perspectives. The first is to study the barriers to innovation in grassroots social governance from the perspective of problem interpretation. From the government level, the government is the leader of grassroots social governance innovation, and grassroots social governance innovation has a strong political nature, and the government pays more attention to efficiency but ignores democracy [5]. It makes the grassroots social governance innovation lack vitality. At the same time, the dislocation and absence of functions of grass-roots governments, insufficient supply of resources, and lagging management reduce the innovation performance to some extent [6]. From the perspective of the public, grassroots social governance innovation is easy to be covered up by technocracy and exclusivity, and the public is rarely the core of previous deliberation, decision-making and development [7], and there is no institutional guarantee for public participation in grassroots social governance innovation. From the perspective of innovation itself, grassroots social governance innovation is not sustainable, innovation subjects lack enthusiasm, innovation projects lack operability, and the use of innovative technologies is insufficient [8]. The second is to study the influencing factors of grassroots social governance innovation from the perspective of causality. Based on the political dimension, some scholars have discussed the impact of government authority, political support [9], organizational structure, power structure, legal structure, action structure, grassroots social governance system, and the "trinity" model on grassroots social governance innovation [10]. Other scholars have studied the interactions between personal attributes, political leaders and their environment [11], top-down political mandates and policy pressures, civil rights and civic participation [12], and the overlap of innovative activities with existing social structures and social needs [13]. The net effect of other factors on grassroots social governance innovation. The third is to study the innovation path of grassroots social governance based on the goal-oriented perspective. From the government level, the government should optimize the grid management system in governance innovation, build the network structure of actors, and form innovation synergy [14]. The government should further integrate various resources, respect the interests of multiple subjects, strengthen political commitment, optimize the collaborative governance model, improve the grassroots governance innovation system, and form a grassroots governance innovation community. From the public level, improve the enthusiasm of the public to participate in grassroots social governance innovation, improve the "government-society" cooperation mechanism, strengthen technological innovation and application, and stimulate the vitality of grassroots social governance innovation [15];From the perspective of the innovation process, the innovation process of grassroots social governance itself is conducive to fostering a more general democracy, which requires strengthening the use of technology, coordinating the interests of all parties, and improving infrastructure [16]. The innovation process needs to be more open to public scrutiny and wider participation.

Scholars have explored the innovation of grassroots social governance from different perspectives, which is of reference significance to this paper. But there are two limitations to the existing research: First, most researchers only study the innovation performance of grassroots social governance from a single factor, and cannot clearly identify the "interactive relationship" between different factors. This situation often leads to excessive dependence on a single factor in the innovation performance of grassroots social governance, and it is impossible to

maximize the efficiency of resource utilization through reasonable allocation of resources. Second, few studies have explored the joint effects of multiple factors, nor have in-depth studies been conducted on the anthems of the differences in grassroots social governance innovation performance, the lack of analysis on the interaction mechanism of multiple complex conditions from the perspective of configuration, the configuration law of conditions is not clear, and the differentiation path and combination of improving grassroots social governance innovation performance need to be studied. Therefore, the second goal that this article addresses is whether and to what extent multiple factors are necessary conditions for achieving high-performance grassroots social governance innovation. How do multiple factors couple and interact to enhance grassroots social governance innovation performance?

Based on this, the main contributions of this paper are as follows: First of all, from the perspective of configuration, this paper discusses the impact of coordination and integration of multiple factors on the innovation performance of grassroots social governance, expands the study based on single variable "net effect" to the study of multi-factor comprehensive effect based on the perspective of configuration, and clarifies the impact of each condition on the innovation performance of grassroots social governance and the linkage interaction between conditions. Secondly, the TOE framework is introduced in this paper to explore the influencing factors of grassroots social governance innovation performance from three aspects: technology, organization and environment, which is conducive to building a multidimensional comprehensive governance model of grassroots social governance innovation. Finally, this paper conducts a comprehensive configuration analysis to explore the optimal configuration of the high innovation performance path in grassroots social governance. It reveals the core influencing factors and the substitution effect among different paths, demonstrating that there is not a singular mode for enhancing innovation performance in grassroots social governance. Moreover, it offers novel insights for managers to selectively adopt innovation modes and enhance the innovation performance of grassroots social governance.

## 2 Theory and analysis framework

This paper uses TOE theory (Technology-organization-Environment), TOE theoretical framework was developed by Tornatzky and Fleischer in 1990 and was originally used to analyze the factors affecting the adoption of innovative technologies by enterprises. It is essentially a comprehensive analysis framework based on the Technology application context [17]. TOE theory holds that the application of a new technology is influenced by three factors: technology, organization and environment. As a system analysis framework based on technology application, TOE theoretical framework is used to explore the mechanism of technology to achieve effects in multi-level application scenarios. Since TOE theoretical framework does not point out the specific explanatory variables of technology, organization and environment, scholars can adjust them appropriately, so TOE theoretical framework has strong flexibility and can explain the causes and influencing factors of complex social phenomena well. Grassroots social governance innovation is affected by many factors, which also applies to TOE theoretical framework. The TOE theoretical framework provides a good theoretical perspective for studying the innovation performance of grassroots social governance.

According to the TOE framework, the implementation of technological innovation in enterprises is mainly influenced by technological factors [18], organizational factors [19], and environmental factors [20]. Technological factors refer to the compatibility between technology and organization [21], which provides support and accessibility for organizational innovation, and brings potential benefits to organizations during the innovation process. Organizational factors include organizational structure and size, organizational systems and

culture, and organizational resources and relationships. Environmental factors refer to external pressures and institutional support for technological innovation.

## 2.1 Technical factors and grassroots social governance innovation performance

In order to achieve higher performance in grassroots social governance innovation, it is necessary to break through the problem of insufficient use of technology in the traditional governance model, which requires increasing technological innovation efforts and improving technological innovation ability. Especially with the rapid development of network information technology, governance technology has become an important tool to promote grassroots social governance innovation, and digital technology has become a key driving force to improve grassroots social governance innovation performance [22]. Based on the actual situation of China's grass-roots social governance, China's grass-roots population is large, grass-roots problems are complex, and it is more necessary to promote technological innovation and increase the application of technology, so as to improve the innovation performance of grass-roots social governance. The academic community largely approaches the forefront and hot topics of grassroots social governance from perspectives such as input-output and cost-benefit [23], focusing on whether the application of technology can reduce governance costs and whether technological empowerment can promote performance. However, the application of technology in grassroots social governance relies on the support of technological innovation capabilities and technological infrastructure. Generally speaking, technological innovation capability is the core competitiveness of regional economic development and grassroots governance, and it represents the breadth and depth of grassroots social governance innovation, while technological facility platform is an important form of technology application, and an important tool to facilitate grassroots social governance and improve grassroots social governance performance [24].

## 2.2 Organizational factors and grassroots social governance innovation performance

Grassroots social governance innovation behavior is the specific implementation of strategic decision-making, and grassroots social governance innovation strategic decision-making is decided by the government organization to a large extent, and will be promoted by the superior government, their own degree of attention and government levels. Under the Chinese political system, the willingness of grassroots governments to innovate and develop depends to a certain extent on the financial resources and policy support given by higher level governments. Top-down political pressure is an important driver of innovation in public sector governance [25]. Therefore, grassroots social governance innovation cannot be achieved without the support of the higher level government, and the capital investment, policy inclination and resource allocation of the higher level government are the effective support for achieving grassroots social governance innovation [26]. Local governments' self-importance is the internal driving force for promoting innovation in grassroots social governance. Existing studies start with institutional analysis and development framework, and depict the relationship between the government's self-importance and innovation performance in terms of rule supply, policy formulation, policy implementation, and policy supervision [27]. In China's bureaucratic administrative management system, lower-level governments are controlled by higher-level governments, and governments at different levels have different control effectiveness over authoritative resources. Therefore, government levels largely determine the success or failure and effectiveness of grassroots social governance innovation.

## 2.3 Environmental factors and grassroots social governance performance

Grassroots social governance innovation is inevitably influenced by environmental factors. Since organizations are embedded in the social environment, their basic goal is to adapt, survive and develop their institutional environment to adapt to the changing environment [28]. Based on institutionalism, peer and superior pressure, public pressure, and competitive pressure significantly affect governance performance and innovation level [29]. Some studies have shown that grassroots social governance innovation will be affected by the pressure of inter-governmental competition. In innovation activities, the higher the innovation level of neighboring governments, the greater the pressure on them to promote their own innovation practices, which will lead to the strengthening of innovation practices and improvement of innovation level of governments in the region [30]. In China, local governments respond to public demands in accordance with the requirements of the central government under the pressure of assessment, and the public puts pressure on the government based on efficiently solving the problems of grassroots governance and safeguarding their legitimate rights and interests. The pressure of public demand urges government departments to adopt innovative strategies to improve the performance of grassroots social governance [9]. Open government can be seen as a form of innovation in the public sector, as it requires the government to adopt new technologies with an open attitude [31]. Therefore, open government is an important guarantee for promoting innovation in grassroots social governance.

In summary, based on TOE theoretical framework and existing studies, and combined with the specific context of grassroots social governance innovation performance, this paper selects eight key variables, It selects the technological innovation ability and technical facility support at the technical dimension, the promotion by higher government, government attention level and level of government at the organizational dimension, and the inter-provincial competitive pressure, the public demand pressure and the internal environment openness at the environmental dimension (See Fig 1). Expanding existing studies from the three dimensions of technical factors, organizational factors and environmental factors, focusing on the single factor influence of grassroots social governance innovation performance, ignoring the lack of research on the combination of multiple factors, and then revealing the core influencing factors of grassroots social governance innovation to achieve high performance and the substitution effect between different paths. It provides reference for local governments to adopt differentiated innovation paths of grassroots social governance according to local conditions.

## 3. Research methods, variable measurement and data construction

### 3.1 Qualitative comparative analysis method

In this paper, we abandon the traditional binary linear model and try to use fuzzy set qualitative comparative analysis (fsQCA) from the perspective of configuration to analyze the multiple driving path to improve the innovation performance of grassroots social governance. The qualitative comparative analysis method can solve the data analysis of 10–60 small and medium-sized samples, and find out the logical relationship between the innovation performance of grassroots social governance under different matching modes of ante-cause conditions. The fsQCA method has the following considerations:

First, the QCA method combines quantitative analysis and qualitative comparison, adheres to the case-oriented research on combination phenomena, and realizes the in-depth dialogue between the theoretical path configuration of grassroots social governance innovation performance and the specific grassroots government grassroots social governance innovation cases.

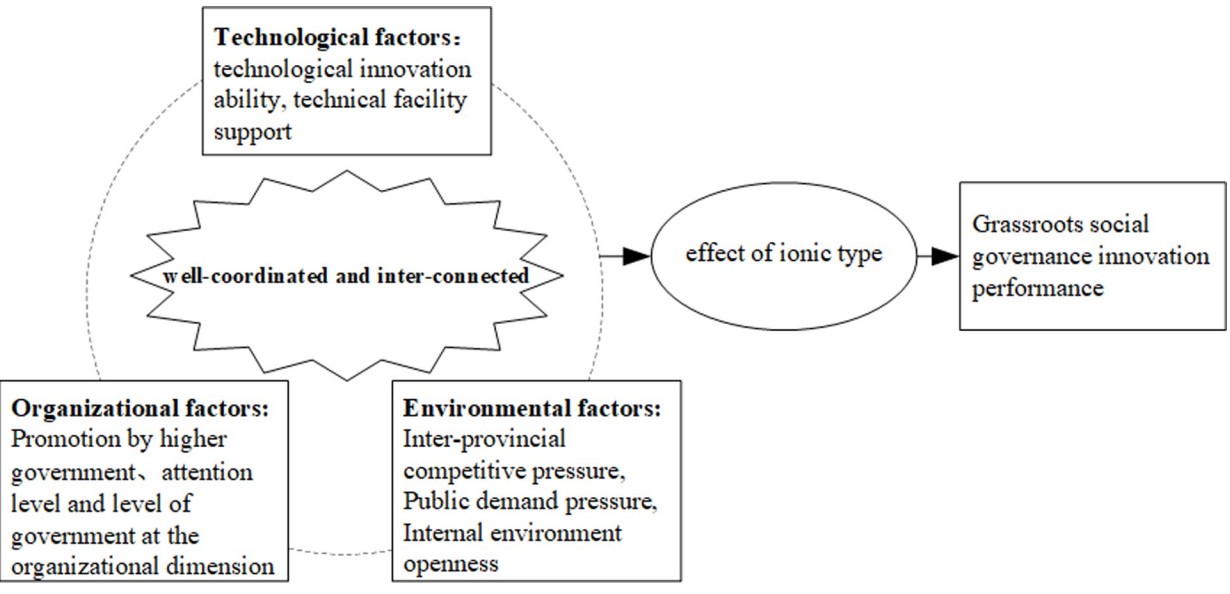

**Fig 1.**

Second, QCA method is not a traditional econometric analysis from the perspective of contingency (studying linear and one-way causality), but based on set theory and from the perspective of configuration, it analyzes the complex mechanism of this phenomenon [32], which can better explain the relationship between different anthems behind the innovation performance of China's grassroots social governance.

Third, compared with configuration research methods such as cluster analysis and factor analysis, the biggest advantage of QCA method is that it can effectively identify dynamic complementarity, configuration equivalence and causal asymmetry among anthems. At the same time, the complex mechanism of grassroots social governance innovation is considered, and most of the influencing factors are continuous variables. Therefore, compared with other QCA methods (clear set qualitative comparative analysis csQCA, multi-value set qualitative comparative analysis mvQCA), fsQCA can more fully capture the subtle changes of anthems at different levels.

In terms of specific technical principles, QCA measures the sufficient and necessary relationship between the antecedent conditional variable X and the result variable Y in the set membership through Consistency and Coverage indexes. Measuring the consistency of set X with respect to Y is a sufficient condition to investigate to what extent X can constitute Y; Measuring the coverage of set X over Y is a necessary condition for examining to what extent X can constitute Y. Coverage describes the explanatory power of interpretation condition (combination) X to result Y. The coverage varies from 0 to 1. The greater the index, the greater the explanatory power of X to Y empirically. The specific calculation formula is as follows:

$$\text{Consistency } (Xi \leq Yi) = \sum[min(Xi, Yi)] / \sum(Xi) \tag{1}$$

$$\text{Coverage } (Xi \leq Yi) = \sum[min(Xi, Yi)] / \sum(Yi) \tag{2}$$

### 3.2 Case selection

The cases selected in this paper are innovative experiments conducted independently by governments at all levels. On the one hand, case selection criteria mainly consider the mutual

application of case selection and condition variable setting, that is, to ensure internal consistency; On the other hand, in order to maintain the balance between the number of cases and the number of conditional variables in QCA research design and general research practices, this paper excluded the influence of some variable factors in case selection, that is, to ensure external consistency. The specific approach is to follow the following criteria in the selection of cases: (1) The standard of universality, which means that the problem situation targeted by the case is universal at the national level; (2) Differentiation criteria: In order to ensure the differentiation between cases, the selected cases cover different fields such as social livelihood, public security prevention and control, community governance, and digital governance; (3) Feasibility criteria: All the innovation cases selected in this paper have been implemented for more than 2 years, which excludes the factors of time and benefit, that is, the selected cases will not have their performance not yet revealed due to time reasons. According to the above case selection criteria, this paper selects 35 grass-roots social governance innovation cases from the "China Top Ten Social Governance Innovation" award selected by the China All-round Well-off Forum and the official news reports of governments at all levels. The data based on the values of each variable come from official data, journal data, and Internet media such as Xinhua-net, NetEase, Sohu and Ifeng.com.

## 3.3 Variable measurement and data construction

**3.3.1 Result variable.** The result variable of this paper is the innovation performance of grassroots social governance. Scholars have divided innovation performance in terms of spatial diffusion [33] or substantive elements [34]. Based on the existing research and the specific practical scenarios of grassroots social governance innovation in China, this paper measures the innovation performance of grassroots social governance as follows:

Firstly, promoting the realization of the value of grassroots social governance and achieving innovation performance in this field is a significant driving force. This can be accomplished by maintaining goal orientation and measuring the innovation performance of grassroots social governance against specific objectives.

Secondly, the replicability and scalability of innovative measures in grassroots social governance are assessed by the introduction of promotion policies by higher-level government authorities and the adoption of these innovative practices by other government departments.

Thirdly, the sustainability of these measures over time can be evaluated based on a two-year timeframe.

Lastly, recognition as an exemplary case is determined by receiving accolades such as the "China's Top Ten Innovations in Social Governance" award and being recognized as a "National Typical Case of Innovative Social Governance in the New Era led by Party Building." The result variable adopts the four-value fuzzy set assignment method, 1 = full membership, 0.67 = partial membership, 0.33 = partial non-membership, 0 = complete non-membership. The result variable meeting one of the conditions and below is 0, meeting two of the conditions is 0.33, meeting three of the conditions is 0.67, and meeting all four is 1, the higher the performance will be. The result variable data comes from government websites, public reports and scholars' studies.

**3.3.2 Conditional variable.** Based on existing studies and typical social governance innovation cases, this paper selects condition variables that may affect the innovation performance of grassroots social governance from three dimensions of technology, organization and environment according to TOE theoretical framework, including 8 condition variables.

*Technological innovation ability*. In this paper, the comprehensive scientific and technological innovation level index is used to measure the technological innovation ability of each

region. The score of comprehensive scientific and technological innovation level index ranges from 0 to 100 points. It is necessary to calibrate the variables of scientific and technological innovation ability in fsQCA, and the key to calibrate is to find out the three qualitative anchor points required for calibration: full membership (1), crossover point (0.5) and complete non-membership (0). Therefore, 100 is 1,50 is 0.5, and 0 is 0. The data comes from the Evaluation Report on China's Regional Innovation Capability (2021) released by the Ministry of Science and Technology.

*Technical facility support*. Scientific and technological support in grassroots social governance innovation is a key factor to guarantee high performance. In this paper, the causal relationship between the technology facility platform and the outcome variable is explored by the measurement criterion of whether to build a technical facility platform. For example, Rizhao built a four-level digital platform, Construction of urban Brain traffic V3.0 in Hangzhou. Therefore, there exists a technical facility platform that provides support for Code 1; otherwise, it is 0. The data comes from government websites, public reports and scholars' researches.

*Promotion by higher government*. In the context of China's bureaucracy, top-down political pressure and financial and policy support of superior governments are important factors to promote the innovation of government organizations. Therefore, this paper mainly examines whether the government's social governance innovation practice is supported by superior leaders or superior policy documents. If it is obtained, the code is 1; otherwise, it is 0. For example, "Jiangsu Huai 'an Public Complaints and Calls Bureau promotes the new model of '126' public complaints and calls work", the General Office of the Provincial Party Committee and the Research Office of the provincial government highly affirm the new model of "126" public complaints and calls work of Huai 'an City, and have published a long survey article in the "Investigation and Research Report" of the provincial Party Committee and the provincial government to make a special recommendation. The variable data comes from government websites, public reports and scholars' studies.

*Government attention level*. As the undertakers and promoters of the innovative practice of grassroots social governance, local government's own importance and preference determine the level of governance performance. Local governments attach importance to the establishment of special governance institutions, guidance of proprietary policies and investment of proprietary funds. If there is construction of special governance institutions or investment of governance policies or investment of governance funds, it is 1; otherwise, it is 0. The variable data comes from government websites, public reports and scholars' studies.

*Level of government*. To some extent, the level of government represents the degree of power and resource superiority. The higher the position of innovation subject in the level of government, the higher the authority of government, and the more formal and informal governance resources available. Therefore, the code of non-grass-roots government above the municipal level (including) is 1, and the code of grass-roots government is 0.

*Inter-provincial competitive pressure*. In this paper, the "China County Social Governance Index Model" developed by the Institute of Social Governance of Zhejiang University is selected to evaluate and rank the government's social governance ability from the four dimensions of social management, co-governance of government and society, social autonomy, and scientific and technological support. If the local government of the case is lower than the neighboring government of the top 100 counties (cities) in social governance, the mean ranking code is 1; otherwise, it is 0. The data comes from the Report on China's Top 100 Counties (cities) in Social Governance.

*Public demand pressure*. The "responsive governance path" based on public demand is an important external driving force for the innovation of grassroots social governance in China. We refer to existing research [15]: If the pressure code of the innovation project is 1 based on

urgency or sudden problems, otherwise it is 0, the variable data comes from government websites, public reports and scholars' researches.

*Internal environment openness.* In the market economy environment, the degree of marketization is highly correlated with the environmental openness inside the government. We refer to existing research [35]: This paper chooses marketization index as the index to measure the environmental openness inside the government. Among them, the marketization index score is 0–10 points. First, find out the qualitative three anchor points required for calibration: full affiliation (1), crossover point (0.5) and complete nonaffiliation (0). Therefore, the marketization index is 1 for 10, 0.5 for 5 and 0 for 0. The data comes from China's Provincial Marketization Index Report (2021) released by the National Economic Research Institute. The measurement and coding of the result and condition variables are shown in Table 1:

## 4 Results and analysis

### 4.1 Analysis of the necessity of individual conditions

Firstly, the qualitative analysis of fuzzy sets should detect whether the conditional variable is a necessary condition for the result. There are two important measurement indexes for necessity detection, namely consistency and coverage. Consistency is equivalent to the degree of significance (p-value test) in regression analysis, that is, the extent to which a certain result requires the presence of a certain variable. It is generally believed that if the consistency reaches 0.9, the necessary condition test criteria are met [36], while the coverage reflects how many cases with a single anthems condition achieve high innovation performance. In this paper, fsQCA3.0 software is used to test whether a single condition constitutes a necessary condition for high innovation performance of grassroots social governance. As can be seen from Table 2, among the eight conditional variables, only the internal environment openness variable reached the consistency testing standard, and the consistency value was 0.9203, that is, 92.03% of the 35 grassroots social governance innovation cases had high internal environment openness. This

**Table 1. Encoding of result variables and condition variables.**

| Variables | Tier 1 Indicator | Secondary Indicator | Variable code |
|---|---|---|---|
| Outcome Variable | Grassroots social governance innovation performance | 1. Achieve governance objectives | Meet one of the conditions and below is 0; Two of them are 0.33; It meets three of the conditions and is 0.67; All four correspond to 1 |
| | | 2. Can be replicated and promoted | |
| | | 3. Sustainable in time | |
| | | 4. Selected excellent cases | |
| Conditional Variable | Technological factors | Technological innovation ability | The scientific and technological innovation level index is 1 out of 100; 50 is 0.5; Zero is zero |
| | | Technical facility support | Platform support is 1; Otherwise zero |
| | Organization factors | Promotion by higher government | Support from superior leaders or superior policy documents is 1; Otherwise zero |
| | | Government attention level | The construction of governance institutions or governance policies or capital input is 1; Otherwise zero |
| | | Level of government | Governments above the county level are 1; Government at the county level and below is 0 |
| | environment factors | Inter-provincial competitive pressure | The average ranking of the top 100 counties (cities) in social governance is 1; Otherwise zero |
| | | Public demand pressure | The pressure of innovation project based on urgency or emergent problem is 1; Otherwise zero |
| | | Internal environment openness | Marketization index 10 is 1; 5 is 0.5; Zero is zero |

**Table 2. The necessity analysis results of high innovation performance in grassroots social governance.**

| Antecedent condition | High innovation performance | |
|---|---|---|
| | Consistency | Coverage |
| Technical innovation ability | 0.8522 | 0.7012 |
| ~ Technical innovation ability | 0.5632 | 0.7206 |
| Technical facility support | 0.6501 | 0.5968 |
| ~ Technical facility support | 0.3567 | 0.5012 |
| Promotion by higher government | 0.6854 | 0.6589 |
| ~ Promotion by higher government | 0.3254 | 0.3542 |
| Government attention level | 0.7596 | 0.6029 |
| ~ Government attention level | 0.2308 | 0.4208 |
| Level of government | 0.5681 | 0.5362 |
| ~ Level of government | 0.3259 | 0.6502 |
| Inter-provincial competitive pressure | 0.3225 | 0.5985 |
| ~ Inter-provincial competitive pressure | 0.6302 | 0.5496 |
| Public demand pressure | 0.4016 | 0.6857 |
| ~ Public demand pressure | 0.6514 | 0.5129 |
| Internal environment openness | 0.9203 | 0.6802 |
| ~ Internal environment openness | 0.3025 | 0.7768 |

Note: "~" indicates not.

also means that grassroots social governance innovation lacking internal environment openness is difficult to achieve high performance. The consistency level of other single conditions on the high innovation performance of grassroots social governance is lower than 0.9, which does not constitute a necessary condition for the result.

## 4.2 Sufficiency analysis of conditional grouping

From the perspective of set theory, configuration analysis is to explore whether the set formed by the combination of antecedent conditions to achieve the high innovation performance of grassroots social governance is a subset of the result set, that is, the adequacy analysis of the results of configuration realization with different antecedent conditions. The adequacy of the configuration is also expressed by consistency. According to the data characteristics, the consistency threshold is set to 0.80 and the acceptable case frequency is set to 1. To avoid inconsistent configurations, set the Proportional Reduction in Inconsistency (PRI) threshold to 0.75 [37], When analyzing the effective configuration that leads to the result, three classes of solutions are generated, namely simple solutions, intermediate solutions and complex solutions. In this paper, by referring to the research of Fiss [32], the intermediate solution is used as the main reference, and through the nested comparison between the intermediate solution and the simple solution, the factors that appear in both the intermediate solution and the simple solution are identified as the core conditions, and the factors that only appear in the intermediate solution are identified as the edge conditions.

As can be seen from Table 3, there are 7 configuration paths for high innovation performance of grassroots social governance, and the overall Solution Consistency is 0.9346, which means that 93.46% of innovation cases of grassroots social governance innovation that meet these 7 types of configuration conditions show a high level of performance. The overall Solution Coverage is 0.6985, which means that the seven-type conditional configuration can explain 69.85% of the high innovation performance cases of grassroots social governance. The

**Table 3. Analysis of configuration results of high innovation performance in grassroots social governance.**

| Antecedent condition | High innovation performance in grassroots social governance | | | | | | |
|---|---|---|---|---|---|---|---|
| | C1 | C2a | C2b | C3a | C3b | C4a | C4b |
| Technological innovation ability | s | | x | s | s | s | s |
| Technical facility support | | S | S | s | s | X | X |
| Promotion by higher government | s | | x | s | s | S | S |
| Government attention level | | S | S | S | S | x | |
| Level of government | | X | X | S | S | S | S |
| Inter-provincial competitive pressure | x | X | X | x | | | s |
| Public demand pressure | s | x | | | x | X | X |
| Internal environment openness | S | S | S | S | S | S | S |
| Consistency | 0.9358 | 0.9685 | 0.9895 | 0.9687 | 0.9802 | 0.9765 | 0.8987 |
| Original coverage | 0.2598 | 0.1265 | 0.0859 | 0.0896 | 0.1108 | 0.0856 | 0.0401 |
| Unique coverage | 0.1865 | 0.1206 | 0.0275 | 0.0058 | 0.0850 | 0.0405 | 0.0412 |
| Global consistency | 0.9346 | | | | | | |
| Overall coverage | 0.6985 | | | | | | |

Note: *S* or *s* Indicates that the condition exists, *X* or *x* Indicates that the condition does not exist, "blank" Indicates that the condition may or may not exist; *S* or *X* Representation core condition, *s* or *x* Representational edge condition. The same below.

consistency of the solution and the coverage of the solution are both higher than the critical value, which indicates that the empirical analysis is effective. Based on the conditional configuration, we can further identify the differentiated adaptive relationship between technology, organization and environment in improving the innovation performance of grassroots social governance. The core conditions of C2a and C2b, C3a and C3b, and C4a and C4b are the same, which constitute the second-order equivalent configuration respectively [32, 38]. Therefore, the seven configurations can be regarded as the combination of sufficient conditions to improve the innovation performance of grassroots social governance, which can be divided into the following four path modes.

**4.2.1 Environment-led technology and organization driven model.** In configuration C1, the openness of internal environment is the core existence condition, the edge existence condition of technological innovation ability, the support of higher government and the pressure of public demand is the edge existence condition, the pressure of intergovernmental competition is the edge non-existence condition, and the other conditions are the uncertain state. It shows that as long as maintaining a high level of environmental openness, improving technological innovation ability, obtaining the support of higher governments, and responding to the public demand, a high level of grassroots social governance innovation performance can be achieved. The consistency level of configuration 1 is 0.9358, indicating that this configuration has a 93.58% possibility to achieve high innovation performance. Its original coverage is 0.2598, indicating that about 25.98% of grassroots social governance innovation cases can explain configuration 1. Its unique coverage is 0.1865, indicating that 18.65% of cases can be explained by this path alone. Typical examples of this model are: Songgang Street, Baoan District, Shenzhen City, Guangdong Province, implements "one grid and multiple" grid grassroots governance. Baoan District, Shenzhen City, Guangdong Province, has a high level of internal environmental openness, the level of marketization and scientific and technological innovation are in the forefront of the country, and ranks 25th in the ranking of the top 100 counties (cities) in social governance, with less pressure of inter-governmental competition. The innovative practice of "one grid and multiple" grid grass-roots governance in Songgang

Street has significantly improved the efficiency of grass-roots social governance. In August 2018, Songgang Street implemented the "one grid multiple" grid governance system in 13 communities under its jurisdiction, sinking personnel, rights and responsibilities to the community line, reducing the work chain, with the gradual improvement of personnel and rights and responsibilities sinking mechanism, its governance system effectively improved the efficiency of work, showing a new atmosphere of grassroots governance.

**4.2.2 Technology, organization and environment driving model.**   This model includes two configurations, C2a and C2b. Configuration C2a indicates that when there is support of technical facilities, more attention to itself and a higher level of openness of the internal environment, even if the government level is low, the pressure of inter-governmental competition and public demand is small, the grassroots social governance innovation will achieve higher performance. This configuration has a 96.85% probability of achieving high innovation performance, and 12.06% of cases can only be explained by this path. A representative case of this path is as follows: Pingshan New District of Shenzhen innovates the "digital governance" model. As a pioneer of China's reform and opening up, Shenzhen has a high level of marketization and openness of the government's internal environment. In the grassroots social governance, we should strengthen the support of technical facilities, build "one library, one team, two networks and two systems" in the software aspect, achieve full coverage of the three-level network of "new district—office—community" in the hardware aspect, and innovate the "Internet + intermediary service" model; In grassroots social governance, the government attaches more importance to further improving relevant laws and regulations, standardizing the service process according to law, strictly implementing evaluation and assessment, and strengthening in-process and post-event supervision. Configuration C2b indicates that when there is technical support, more attention to itself and a higher level of openness of the internal environment, even if the government level is low, the pressure of inter-governmental competition is small, the technological innovation ability is low, and the higher government does not promote, the grassroots social governance innovation will achieve higher performance, but the coverage of this configuration is low, and 2.75% of the cases can only be explained by this path. A representative example of this path is the setting up of a "joint command center" in Tianya District, Sanya City, Hainan Province. As a grass-roots government, Tianya District has little pressure of inter-governmental competition, high environmental openness and government attention level, and builds a joint command center. Although the scientific and technological innovation ability is at a low level in the country, it can still achieve high innovation performance.

**4.2.3 Technology and organization-led environment driven model.**   This model includes two configurations, C3a and C3b, which indicate that under the environment of less pressure of inter-governmental competition or public demand, giving full play to technological advantages such as technological innovation ability, technical facility support, and organizational advantages such as the promotion of higher government, government attention level, and government level, higher innovation performance can be achieved under the drive of internal environment openness. The corresponding cases of C3a configuration mode are as follows: Jinan City of Shandong Province established the "four-society linkage" mechanism, and Jinan City ranks first in the ranking of the top 100 counties (cities) of social governance by neighboring governments, with less pressure of inter-governmental competition. Meanwhile, the openness level of internal environment of Jinan City government ranks above the average level in the country; In terms of technical factors, the technological innovation ability of Jinan city is at the top level of the country, and it creates an information platform to provide information technology support for the "four-social linkage"; At the level of organizational factors, the Ministry of Civil Affairs and the Ministry of Finance put forward the Opinions on

Accelerating Community Social Work Services and the Shandong Provincial Civil Affairs Department issued the Opinions on Promoting the "Four-community Linkage" innovative community governance and services to provide policy support for the "four-community linkage". In addition, the Jinan Municipal Government also attaches great importance to the "four-community linkage". In the "four social linkage", information communication system, team construction system and service joint office system are established. C3b configuration mode corresponding cases such as: The case of "Connected community" governance service new model in the Northern District of Qingdao City, Shandong Province is similar to the case of C3a. In the northern district of Qingdao City, an "connected community" information platform integrating democratic consultation, social situation and public opinion collection, display, service and other functions is fully built to connect residents' needs, government public resources, volunteer service resources and social service resources. It has promoted the deep integration of "Internet plus" and community construction.

**4.2.4 Organization and environment-led technology driven model.** The model includes two configurations, C4a and C4b. In the two configurations, the promotion by the superior government, the openness of the government level and the internal environment are the core existence conditions, the support of technical facilities and the pressure of public demand are the core non-existence conditions, the technological innovation ability is the marginal existence conditions, and the pressure of inter-governmental competition is the marginal existence or uncertainty conditions. Government attention level is non-existence or uncertain conditions at the edge. The results show that in the absence of technical facilities support and public demand pressure, as long as the non-grassroots government receives the support of the higher level government in the open market environment, supplemented by higher technological innovation ability, the grassroots social governance innovation can achieve higher performance regardless of whether there is pressure of inter-governmental competition or whether it attaches importance to it. C4a configuration mode corresponding cases such as: Yangpu District of Shanghai carries out the "Community and Good Neighborliness Center", and the internal environment openness and technological innovation ability of the Shanghai Municipal government rank first in the former country. The government also attaches great importance to the construction of "community and good neighborliness Center", and actively supports and promotes the popularization and promotion of "Community and good neighborliness Center" in Shanghai and even in the whole country. The innovation practice of "Community good-neighborly Center" in Yangpu District of Shanghai has also been selected as the national excellent social governance innovation case for many times. The corresponding cases of C4b configuration mode are as follows: Zhejiang Provincial Civil Affairs Department implements "village rules and people's conventions to promote harmony, social conventions to ensure governance", community governance takes village rules and people's conventions and residents' conventions as the carrier, innovates the credit management system and mechanism of letters and visits, constantly improves the grass-roots governance system combining "rule of law, rule of virtue and autonomy", and gradually improves the innovative performance of grass-roots social governance.

## 4.3 Robustness test

Adjustment of condition variables, adjustment of calibration threshold, change of consistency, change of case frequency, etc., can be used as the robustness test criteria for qualitative comparative analysis, and can also be tested by changing data sources, adjusting measurement methods, and changing data collection period [39], To judge whether the original analysis results are robust, there are two levels of criteria: on the one hand, the fitting parameters of the

re-analysis results are generally considered robust if there is little difference; On the other hand, if there is still a clear subset relationship between the result configuration of the robustness test and the original configuration result, the original result is considered robust [36]. In this paper, the original consistency threshold of the solution is adjusted from 0.8 to 0.85 based on two criteria for robustness evaluation and existing calibration methods [40]. The results of conditional configuration adequacy analysis of the high innovation performance of grassroots social governance obtained after modifying the calibration method are shown in Table 4. After comparison and calculation, it is found that the configuration results under the adjustment strategy are basically consistent with the original configuration results or there is an obvious subset relationship, which further confirms the robustness of the research conclusions.

# 5 Discussion and conclusion

## 5.1 Discussion

Based on the above research result, this paper provides the following management implications for government departments to improve the innovation performance of grassroots social governance:

Firstly, the openness of the internal environment is a crucial and indispensable factor in enhancing the innovation performance of grassroots social governance. In the era of information technology, it is imperative for the government to exhibit a more open and inclusive mindset towards emerging technologies, actively embracing and accommodating new advancements. This entails being responsive, adaptable, and receptive to the opportunities and challenges presented by information technology. It reflects the factors of organizational system culture. From the historical development experience, politics and economy are closely intertwined, whether it is Milton Friedman's promotion of economic freedom to political freedom [41], The mutual support between the openness of power in the economic field and the openness of power in the political field by Douglass C. North both demonstrated a high positive correlation between the openness of environment within the government and the degree of marketization [42]. Therefore, the government should follow the development of the market economy and further improve the degree of marketization.

**Table 4. Robustness test results.**

| Antecedent condition | High innovation performance in grassroots social governance | | | | | |
|---|---|---|---|---|---|---|
| | C1 | C2a | C2b | C3a | C3b | C4 |
| Technological innovation ability | s | | x | s | s | s |
| Technical facility support | | S | S | s | s | X |
| Promotion by higher government | s | | x | | s | S |
| Government attention level | | S | S | S | S | |
| Level of government | | X | X | S | S | S |
| Inter-provincial competitive pressure | x | X | X | x | | |
| Public demand pressure | s | x | | | x | X |
| Internal environment openness | S | S | S | S | S | S |
| Consistency | 0.9358 | 0.9685 | 0.9895 | 0.9687 | 0.9802 | 0.9802 |
| Original coverage | 0.2598 | 0.1265 | 0.0859 | 0.0896 | 0.1108 | 0.1926 |
| Unique coverage | 0.1865 | 0.1206 | 0.0275 | 0.0058 | 0.0850 | 0.1196 |
| Global consistency | 0.9502 | | | | | |
| Overall coverage | 0.5306 | | | | | |

Secondly, there are synergies among technology, organization and environment, which highlights the complexity of improving the innovation performance of grassroots social governance. The second is to study all the important factors that affect innovation performance, focusing on a single factor may not produce the desired results. It is crucial to consider multiple factors from a holistic perspective and adopt context-specific measures that align with the local conditions. This approach acknowledges the interplay and interdependencies among technology, organization, and environment in order to effectively improve innovation performance in grassroots social governance. We will develop differentiated paths for community-level social governance innovation. The government can make efforts from three aspects: First, strengthen the technical support for grassroots social governance innovation. We will apply advanced technologies such as big data, blockchain, Internet of Things, and 5G to multiple scenes of grassroots social governance, and strive for overtaking on corners and leapfrog development. The second is to strengthen the organizational construction of grassroots social governance innovation [43]. In the public sector, especially in government organizations, the willingness of lower-level governments to innovate and develop depends to a certain extent on the financial resources and policy support provided by higher-level governments, which have a strong dependence on higher-level government organizations. Thus, top-down political pressure is the most dominant external driver of applied innovation in the public sector. China's vertical governments at all levels are in essence a "relationship between superiors and subordinates". The policy support of higher governments to lower governments reflects a certain attribute of political pressure, which can bring various resources and pressures to lower governments. Therefore, higher governments should actively encourage and support lower governments to promote innovative practice of grassroots social governance. In addition, studies have shown that local governments' self-importance has a significant positive impact on innovation performance. Local governments should attach importance to grassroots social governance innovation practices and formulate development strategies, norms and measures according to local conditions. The third is to optimize the external environment for grassroots social governance innovation, improve its own competitiveness, actively respond to public demand, build an institutional environment with multi-subject participation, and create a grassroots social governance community with co-construction of subjects, co-governance of resources and shared benefits.

Thirdly, whether considering the entire nation or taking into account regional disparities, there exists not just one "satisfactory solution" for enhancing the innovation performance of grassroots social governance or fostering innovation in grassroots social governance. In other words, a universal, "one-size-fits-all" approach does not exist. Instead, one potential starting point is to prioritize the expansion of the internal environment's openness. It can also promote innovation in social governance at the grassroots level from the aspects of technological innovation capacity, technical facility support, promotion by the higher level government, and their own importance. There are multiple paths to improve innovation performance, not a single one. Therefore, local governments should select appropriate paths and targeted measures based on local development scenarios and governance resources, and find an efficient way to innovate grassroots social governance in China.

This paper possesses certain limitations and suggests avenues for future research. Firstly, it focuses solely on preconditions such as technology, organization, and environment, while there are numerous other factors that impact innovation performance. To achieve a more comprehensive understanding of the influencing factors, future research could incorporate additional elements such as strategic orientation and organizational resources. Secondly, this study selected 35 grassroots social governance innovation cases, and the generalizability of the qualitative comparative analysis results may be restricted by the sample size. Collecting a larger

dataset for further analysis in the future would enhance the robustness of the findings. As a result, this study holds significant theoretical and practical value, offering fresh ideas and methodologies for the field of management and social governance research. It opens up new avenues for exploration and provides directions to advance knowledge in these domains.

## 5.2 Conclusion

Promoting innovation practices in grassroots social governance and enhancing the performance of grassroots social governance innovation is a key driver for modernizing the grassroots social governance system and governance capacity. In the current Chinese political context, local governments promote grassroots social governance innovation plans in a timely and effective manner, with varying degrees of success. Why do these differences exist, and how do various factors interact to enhance grassroots social governance innovation performance? Based on this, this paper constructs 35 grassroots social governance innovation case data and uses the TOE analysis framework and innovative fuzzy set qualitative comparative analysis (fsQCA) method to explore the "combined effects" and "interactive relationships" of eight antecedent conditions—technological innovation ability, Technical facility support, Promotion by higher government, self-value, government level, inter-facility competition pressure, public demand pressure, and internal environmental openness—on high innovation performance in grassroots social governance. The specific research conclusions are as follows: First, the openness of the internal environment is a necessary condition for improving the innovation performance of grassroots social governance, forming a single-factor path of high innovation performance. Second, achieving high innovation performance in grassroots social governance has the characteristics of "multiple paths and the same goal", with multiple different antecedent conditions forming 7 paths that ultimately achieve the goal of high innovation performance, namely the same goal. The seven configuration paths can be summarized as environment-led technology and organization-driven mode, technology, organization and environment drive mode, technology and organization-led environment drive mode, and organization and environment-led technology drive mode. Third, from the perspective of the combination conditions of the high innovation performance of grassroots social governance, the influencing factors of the high innovation performance of grassroots social governance are the result of the joint action of technology, organization, environment and other factors, which means that the dynamic mechanism to improve the innovation performance of grassroots social governance is often mixed, rather than a specific factor that can be imposed and promoted.

## Supporting information

**S1 Data.**
(XLS)

## Acknowledgments

This research benefited from the comments and feedback received on the findings presented in seminars held at the East China University of Political Science and Law, Shanghai Jiaotong University. Our particular thanks go to Ruisong Quan, Changxu Zheng, and the other members of our project advisory team.

## Author Contributions

**Conceptualization:** Longshun Xu, Huange Xu.

**Data curation:** Longshun Xu.

**Formal analysis:** Longshun Xu.

**Software:** Huange Xu.

**Visualization:** Huange Xu.

**Writing – original draft:** Shuoliang Jiang.

**Writing – review & editing:** Nana Song, Xiansheng Chen, Shuoliang Jiang.

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
