## [Decision Letter · Decision Letter 0]

13 Sep 2023

PONE-D-23-21194How to achieve higher performance in grassroots social governance innovation in China？——fsQCA based on TOE framework

PLOS ONE

Dear Dr. Chen,

Thank you for submitting your manuscript to PLOS ONE. After careful consideration, we feel that it has merit but does not fully meet PLOS ONE’s publication criteria as it currently stands. Therefore, we invite you to submit a revised version of the manuscript that addresses the points raised during the review process.

The two reviewers provided valuable suggestions. I hope it will be helpful for your article to be published, and the following are the issues that I am particularly concerned about. If they cannot be resolved, I cannot recommend acceptance.

1. Lines 26-41 are missing references.2. 1. Lines 91-99 are missing references.3. The discussion section is missing references. And the conclusion should be discussed below.4. There are too few and old references.

We look forward to receiving your revised manuscript.

Kind regards,

Chunyu Zhang

Academic Editor

PLOS ONE

“Funding

Youth Project of Humanities and Social Sciences Foundation of Ministry of Education: Research on the Mechanism and Policy Effect of Cross-basin Horizontal Ecological Compensation on Enterprises' High-quality Development (grant number 22YJC630172)

Partial financial support was received from the Shanghai Philosophy and Social Science Planning Fund Project: "Pudong New Area to build a Leading area of Socialist Modernization Construction" (grant number 2021BZZ002).”

Reviewers' comments:

Reviewer's Responses to Questions

**Comments to the Author**

1. Is the manuscript technically sound, and do the data support the conclusions?

Reviewer #1: Yes

Reviewer #2: Yes

2. Has the statistical analysis been performed appropriately and rigorously? 

Reviewer #1: Yes

Reviewer #2: Yes

3. Have the authors made all data underlying the findings in their manuscript fully available?

Reviewer #1: Yes

Reviewer #2: Yes

4. Is the manuscript presented in an intelligible fashion and written in standard English?

Reviewer #1: Yes

Reviewer #2: Yes

5. Review Comments to the Author

Reviewer #1: In China, grassroots social governance innovation is a very fashionable research topic, but many studies are vague about the concept of grassroots society or grassroots, and such studies lack authenticity. The article does not clearly define and discuss the concept of grassroots society. Suggest the author to add.

Reviewer #2: At present, social governance innovation is an important issue. Although this study has achieved some results, however, some problems still remain. I put forward some suggestions for the author's reference.

1.Theory and analysis framework: This part should be further improved. In particular, the development and current status of the theory, as well as the elaboration of current related research.

2.The format of 320-334、355-359 and line376-382 should be adjusted.

3. The Discussion should be further enriched. The current Discussion is not enlightening. In addition to the discussion of the results, there should be an extension of the author's thinking to enlighten the reader. This is one of the most important things in the manuscript.

4. Increased emphasis on research contributions/values, limitations and future research directions.

5. References must be further increased, especially those within the last three years.

6. PLOS authors have the option to publish the peer review history of their article (what does this mean?). If published, this will include your full peer review and any attached files.

Reviewer #1: **Yes: **liu bangfan

Reviewer #2: No

---

## [Author Response · Author response to Decision Letter 0]

21 Oct 2023

Dear Academic Editor Zhang, 

We are very grateful for your consideration and suggestions about our manuscript entitled “How to achieve higher performance in grassroots social governance innovation in China？——fsQCA based on TOE framework” (Reference No: PONE-D-23-21194). We worked really hard on this article, which is related to whether a doctor can graduate on time. We have integrated the comments into the revised manuscript, and now we enclose the revised manuscript for your consideration for publication in PLOS ONE.

The followings are our responses with red fonts to the comments of the reviewers. After each the comments from the reviewers that are in italics, our answers are given. In addition, We updated the title logic, updated the cases and data, asked English professionals to polish the article, and highlighted it in color.

Best wishes,

Xiansheng Chen

 Academic Editor Zhang #:

1. Lines 26-41 are missing references.

Answer: We agree with this point. We have added the following content and references in line 26-47 based on the comments of reviewer #1:

Grassroots social governance has attracted wide attention[1]. Grassroots society refers to the lowest and fundamental level of social organization and activities. It typically refers to the basic social units composed of residents, communities, villages, streets, etc., in urban or rural areas[2]. Grassroots society is characterized by direct engagement with residents, serving them, and addressing their practical issues[3].

1. Raj G, Feola G, Hajer M, Runhaar H. Power and empowerment of grassroots innovations for sustainability transitions: A review. Environ Innov Soc Tr. 2022;43:375-92. http://doi.org/10.1016/j.eist.2022.04.009

 2. Lv L, Shi D. Innovative Development and Practice of Digital Rural Governance Model Based on Green Ecology. Sustainability-Basel. 2023;15(4). http://doi.org/10.3390/su15042955

 3. Yang Y, Wu F. The Sustainability of the Project-Driven Innovation of Grassroots Governance: Influencing Factors and Combination Paths. Sustainability-Basel. 2022;14(24). http://doi.org/10.3390/su142416862

 4. Li J, Zhan G, Dai X, Qi M, Liu B. Innovation and Optimization Logic of Grassroots Digital Governance in China under Digital Empowerment and Digital Sustainability. Sustainability-Basel. 2022;14(24). http://doi.org/10.3390/su142416470

 5. Farid M. Advocacy in Action: China's Grassroots NGOs as Catalysts for Policy Innovation. Stud Comp Int Dev. 2019;54(4):528-49. http://doi.org/10.1007/s12116-019-09292-3

 6. Qin Y. Grassroots governance and social development: theoretical and comparative legal aspects. Humanities & Social Sci Communications. 2023;10(1). http://doi.org/10.1057/s41599-023-01830-8

2. Lines 91-99 are missing references.

Answer: We agree. We updated the content on lines 96-104 and added the following references:

In the context of the TOE framework, information technology and platforms are considered important factors in enhancing innovation intensity[4]. Scientific and technological innovation can break down inherent institutional barriers between organizations and strengthen communication between participants through digital information systems or technology resource sharing platforms, thereby reducing information asymmetry and increasing the support capacity of information technology[30]. Organizational factors are key variables that affect technological innovation, and larger organizations tend to have advantages in resource allocation and innovation capabilities[31]. External pressure from government competitors and customers, as well as institutional cultural support, are also important factors influencing technological innovation[32].

4. Li J, Zhan G, Dai X, Qi M, Liu B. Innovation and Optimization Logic of Grassroots Digital Governance in China under Digital Empowerment and Digital Sustainability. Sustainability-Basel. 2022;14(24). http://doi.org/10.3390/su142416470

30. Shi Z, Wu YJ, Chiu YH, Chang TH. Research on the influence of technological innovation and technological application: Evidence from China. J Eng Technol Manage. 2022;63. http://doi.org/10.1016/j.jengtecman.2021.101670

31. Sillig C. The role of ideology in grassroots innovation: An application of the arenas of development framework to organic in Europe. Ecol Econ. 2022;191. http://doi.org/10.1016/j.ecolecon.2021.107252

32. Mamonov S, Peterson R. The role of IT in organizational innovation-A systematic literature review. J Strategic Inf Syst. 2021;30(4). http://doi.org/10.1016/j.jsis.2021.101696

3. The discussion section is missing references. And the conclusion should be discussed below.

Answer: Thanks for your suggestion, we have adjusted the position of the conclusion and discussion section, enriched the discussion content and updated the references：

Based on the above research conclusions, this paper provides the following management implications for government departments to improve the innovation performance of grassroots social governance:

First, the openness of the internal environment is an indispensable key factor to improve the innovation performance of grassroots social governance. In the information age, the openness of the internal environment of the government requires it to hold a more open and inclusive attitude towards new things including information technology, and be able to flexibly perceive, actively respond to and accept the opportunities and challenges that information technology may bring[44]. It reflects the factors of organizational system culture. From the historical development experience, politics and economy are closely intertwined, whether it is Milton Friedman's promotion of economic freedom to political freedom [45], The mutual support between the openness of power in the economic field and the openness of power in the political field by Douglass C. North both demonstrated a high positive correlation between the openness of environment within the government and the degree of marketization[46]. Therefore, the government should follow the development of the market economy and further improve the degree of marketization.

Second, the existence of concurrent synergies among technology, organization and environment reveals the complexity of improving innovation performance of grassroots social governance. Although technology, organization and environment are all important factors affecting innovation performance, the focus on a single factor is difficult to achieve the expected effect, and the adaptation of multiple factors should be paid attention to from an "overall perspective" and corresponding measures should be adopted "according to local conditions"[47]. We will develop differentiated paths for community-level social governance innovation. The government can make efforts from three aspects: First, strengthen the technical support for grassroots social governance innovation[48]. We will apply advanced technologies such as big data, blockchain, Internet of Things, and 5G to multiple scenes of grassroots social governance, and strive for overtaking on corners and leapfrog development. The second is to strengthen the organizational construction of grassroots social governance innovation[49]. In the public sector, especially in government organizations, the willingness of lower-level governments to innovate and develop depends to a certain extent on the financial resources and policy support provided by higher-level governments, which have a strong dependence on higher-level government organizations. Thus, top-down political pressure is the most dominant external driver of applied innovation in the public sector[50]. China's vertical governments at all levels are in essence a "relationship between superiors and subordinates". The policy support of higher governments to lower governments reflects a certain attribute of political pressure, which can bring various resources and pressures to lower governments. Therefore, higher governments should actively encourage and support lower governments to promote innovative practice of grassroots social governance. In addition, studies have shown that local governments' self-importance has a significant positive impact on innovation performance. Local governments should attach importance to grassroots social governance innovation practices and formulate development strategies, norms and measures according to local conditions[51]. The third is to optimize the external environment for grassroots social governance innovation, improve its own competitiveness, actively respond to public demand, build an institutional environment with multi-subject participation, and create a grassroots social governance community with co-construction of subjects, co-governance of resources and shared benefits[52].

Third, whether from the perspective of the whole country or from the perspective of regional differences, there is more than one "satisfactory solution" for the improvement of grassroots social governance innovation performance or grassroots social governance innovation, that is to say, there is no "one-size-fits-all" generation path, which can be started from expanding the openness of the internal environment[52]. It can also promote innovation in social governance at the grassroots level from the aspects of technological innovation capacity, technical facility support, promotion by the higher level government, and their own importance. There are multiple paths to improve innovation performance, not a single one[54]. Therefore, local governments should select appropriate paths and targeted measures based on local development scenarios and governance resources, and find an efficient way to innovate grassroots social governance in China.

This paper has the following shortcomings and provides directions for future research: firstly, this paper only considers preconditions such as technology, organisation and environment, but there are many factors that affect innovation performance. Future research can incorporate factors such as strategic orientation and organisational resources to study the influencing factors of innovation performance more comprehensively. Second, 35 grassroots social governance innovation cases were selected for this paper, and the generalisability of the results of the qualitative comparative analysis in terms of application is limited by the sample size. In the future, more data can be collected for further analysis.

44. Pang SL, Dou ST, Li H. Synergy effect of science and technology policies on innovation: Evidence from China. Plos One. 2020;15(10). http://doi.org/10.1371/journal.pone.0240515

45. Turner R. Un/inhibited organism: Political power, psychophysiology, and techniques.; 2009.

46. Wang W, Wen J, Luo ZG, Luo WY. How does environmental punishment affect regional green technology innovation?-Evidence from Chinese Provinces. Plos One. 2023;18(7). http://doi.org/10.1371/journal.pone.0288080

47. Liang LW, Wang ZB, Luo D, Wei Y, Sun JW. Synergy effects and it's influencing factors of China's high technological innovation and regional economy. Plos One. 2020;15(5). http://doi.org/10.1371/journal.pone.0231335

48. Yang HC, Li LS, Liu YB. The effect of manufacturing intelligence on green innovation performance in China. Technol Forecast Soc. 2022;178. http://doi.org/10.1016/j.techfore.2022.121569

49. Rehman SU, Ashfaq K, Bresciani S, Giacosa E, Mueller J. Nexus among intellectual capital, interorganizational learning, industrial Internet of things technology and innovation performance: a resource-based perspective. J Intellect Cap. 2023;24(2):509-34. http://doi.org/10.1108/JIC-03-2021-0095

50. Andersen SC, Jakobsen ML. Political Pressure, Conformity Pressure, and Performance Information as Drivers of Public Sector Innovation Adoption. Int Public Manag J. 2018;21(2):213-42. http://doi.org/10.1080/10967494.2018.1425227

51. He C, Wang YP, Tang K. Impact of Low-Carbon City Construction Policy on Green Innovation Performance in China. Emerg Mark Financ Tr. 2023;59(1):15-26. http://doi.org/10.1080/1540496X.2022.2089019

52. Xu JH, He SY. Can Grid Governance Fix the Party-state's Broken Windows? A Study of Stability Maintenance in Grassroots China. China Quart. 2022;251:843-65. http://doi.org/10.1017/S0305741022000509

53. Bianchi I. Empowering policies for grassroots welfare initiatives: Blending social innovation and commons theory. Eur Urban Reg Stud. 2023;30(2):107-20. http://doi.org/10.1177/09697764221129532

54. Qian X, Cai Y, Yin C. Driving Force of Grassroots Self-governance in Beijing's Neighborhoods: Social Capital, Community Network and Community Service Motivation. Lex Localis. 2019;17(1):159-77. http://doi.org/10.4335/17.1.159-177(2019)

4. There are too few and old references.

Answer: Thanks for your suggestion, we have updated the content and references

Response to Reviewer #1:

In China, grassroots social governance innovation is a very fashionable research topic, but many studies are vague about the concept of grassroots society or grassroots, and such studies lack authenticity. The article does not clearly define and discuss the concept of grassroots society. Suggest the author to add.

Answer: We agree with this point. In lines 27-31, we have added an explanation about grassroots society:

 “Grassroots social governance has attracted wide attention. Grassroots society refers to the lowest and fundamental level of social organization and activities. It typically refers to the basic social units composed of residents, communities, villages, streets, etc., in urban or rural areas. Grassroots society is characterized by direct engagement with residents, serving them, and addressing their practical issues.”

1. Raj G, Feola G, Hajer M, Runhaar H. Power and empowerment of grassroots innovations for sustainability transitions: A review. Environ Innov Soc Tr. 2022;43:375-92. http://doi.org/10.1016/j.eist.2022.04.009

 2. Lv L, Shi D. Innovative Development and Practice of Digital Rural Governance Model Based on Green Ecology. Sustainability-Basel. 2023;15(4). http://doi.org/10.3390/su15042955

 3. Yang Y, Wu F. The Sustainability of the Project-Driven Innovation of Grassroots Governance: Influencing Factors and Combination Paths. Sustainability-Basel. 2022;14(24). http://doi.org/10.3390/su142416862

 4. Li J, Zhan G, Dai X, Qi M, Liu B. Innovation and Optimization Logic of Grassroots Digital Governance in China under Digital Empowerment and Digital Sustainability. Sustainability-Basel. 2022;14(24). http://doi.org/10.3390/su142416470

 5. Farid M. Advocacy in Action: China's Grassroots NGOs as Catalysts for Policy Innovation. Stud Comp Int Dev. 2019;54(4):528-49. http://doi.org/10.1007/s12116-019-09292-3

 6. Qin Y. Grassroots governance and social development: theoretical and comparative legal aspects. Humanities & Social Sci Communications. 2023;10(1). http://doi.org/10.1057/s41599-023-01830-8

Response to Reviewer #2:

At present, social governance innovation is an important issue. Although this study has achieved some results, however, some problems still remain. I put forward some suggestions for the author's reference.

1.Theory and analysis framework: This part should be further improved. In particular, the development and current status of the theory, as well as the elaboration of current related research.

Answer: We agree, In the theoretical part, we added the following contents and references, and systematically expounded the research status：

TOE theoretical framework (Technology-organization-Environment) was developed by Tornatzky and Fleischer in 1990 and was originally used to analyze the factors affecting the adoption of innovative technologies by enterprises. It is essentially a comprehensive analysis framework based on the Technology application context[24].

In the context of the TOE framework, information technology and platforms are considered important factors in enhancing innovation intensity[4]. Scientific and technological innovation can break down inherent institutional barriers between organizations and strengthen communication between participants through digital information systems or technology resource sharing platforms, thereby reducing information asymmetry and increasing the support capacity of information technology[30]. Organizational factors are key variables that affect technological innovation, and larger organizations tend to have advantages in resource allocation and innovation capabilities[31]. External pressure from government competitors and customers, as well as institutional cultural support, are also important factors influencing technological innovation[32].

4. Li J, Zhan G, Dai X, Qi M, Liu B. Innovation and Optimization Logic of Grassroots Digital Governance in China under Digital Empowerment and Digital Sustainability. Sustainability-Basel. 2022;14(24). http://doi.org/10.3390/su142416470

24. Tornatzky LG, Fleischer M, Chakrabarti AK. Processes of technological innovation.: Lexington books; 1990.

30. Shi Z, Wu YJ, Chiu YH, Chang TH. Research on the influence of technological innovation and technological application: Evidence from China. J Eng Technol Manage. 2022;63. http://doi.org/10.1016/j.jengtecman.2021.101670

31. Sillig C. The role of ideology in grassroots innovation: An application of the arenas of development framework to organic in Europe. Ecol Econ. 2022;191. http://doi.org/10.1016/j.ecolecon.2021.107252

32. Mamonov S, Peterson R. The role of IT in organizational innovation-A systematic literature review. J Strategic Inf Syst. 2021;30(4). http://doi.org/10.1016/j.jsis.2021.101696

2.The format of 320-334、355-359 and line376-382 should be adjusted.

Answer: Thank you for your suggestion, We readjusted the paragraphs and spacing and content.

3. The Discussion should be further enriched. The current Discussion is not enlightening. In addition to the discussion of the results, there should be an extension of the author's thinking to enlighten the reader. This is one of the most important things in the manuscript.

Answer: We agree. We have enriched the discussion section and added references. The specific changes and references are as follows:

5.1 Discussion

Based on the above research conclusions, this paper provides the following management implications for government departments to improve the innovation performance of grassroots social governance:

First, the openness of the internal environment is an indispensable key factor to improve the innovation performance of grassroots social governance. In the information age, the openness of the internal environment of the government requires it to hold a more open and inclusive attitude towards new things including information technology, and be able to flexibly perceive, actively respond to and accept the opportunities and challenges that information technology may bring[44]. It reflects the factors of organizational system culture. From the historical development experience, politics and economy are closely intertwined, whether it is Milton Friedman's promotion of economic freedom to political freedom [45], The mutual support between the openness of power in the economic field and the openness of power in the political field by Douglass C. North both demonstrated a high positive correlation between the openness of environment within the government and the degree of marketization[46]. Therefore, the government should follow the development of the market economy and further improve the degree of marketization.

Second, the existence of concurrent synergies among technology, organization and environment reveals the complexity of improving innovation performance of grassroots social governance. Although technology, organization and environment are all important factors affecting innovation performance, the focus on a single factor is difficult to achieve the expected effect, and the adaptation of multiple factors should be paid attention to from an "overall perspective" and corresponding measures should be adopted "according to local conditions"[47]. We will develop differentiated paths for community-level social governance innovation. The government can make efforts from three aspects: First, strengthen the technical support for grassroots social governance innovation[48]. We will apply advanced technologies such as big data, blockchain, Internet of Things, and 5G to multiple scenes of grassroots social governance, and strive for overtaking on corners and leapfrog development. The second is to strengthen the organizational construction of grassroots social governance innovation[49]. In the public sector, especially in government organizations, the willingness of lower-level governments to innovate and develop depends to a certain extent on the financial resources and policy support provided by higher-level governments, which have a strong dependence on higher-level government organizations. Thus, top-down political pressure is the most dominant external driver of applied innovation in the public sector[50]. China's vertical governments at all levels are in essence a "relationship between superiors and subordinates". The policy support of higher governments to lower governments reflects a certain attribute of political pressure, which can bring various resources and pressures to lower governments. Therefore, higher governments should actively encourage and support lower governments to promote innovative practice of grassroots social governance. In addition, studies have shown that local governments' self-importance has a significant positive impact on innovation performance. Local governments should attach importance to grassroots social governance innovation practices and formulate development strategies, norms and measures according to local conditions[51]. The third is to optimize the external environment for grassroots social governance innovation, improve its own competitiveness, actively respond to public demand, build an institutional environment with multi-subject participation, and create a grassroots social governance community with co-construction of subjects, co-governance of resources and shared benefits[52].

Third, whether from the perspective of the whole country or from the perspective of regional differences, there is more than one "satisfactory solution" for the improvement of grassroots social governance innovation performance or grassroots social governance innovation, that is to say, there is no "one-size-fits-all" generation path, which can be started from expanding the openness of the internal environment[52]. It can also promote innovation in social governance at the grassroots level from the aspects of technological innovation capacity, technical facility support, promotion by the higher level government, and their own importance. There are multiple paths to improve innovation performance, not a single one[54]. Therefore, local governments should select appropriate paths and targeted measures based on local development scenarios and governance resources, and find an efficient way to innovate grassroots social governance in China.

This paper has the following shortcomings and provides directions for future research: firstly, this paper only considers preconditions such as technology, organisation and environment, but there are many factors that affect innovation performance. Future research can incorporate factors such as strategic orientation and organisational resources to study the influencing factors of innovation performance more comprehensively. Second, 35 grassroots social governance innovation cases were selected for this paper, and the generalisability of the results of the qualitative comparative analysis in terms of application is limited by the sample size. In the future, more data can be collected for further analysis.

44. Pang SL, Dou ST, Li H. Synergy effect of science and technology policies on innovation: Evidence from China. Plos One. 2020;15(10). http://doi.org/10.1371/journal.pone.0240515

45. Turner R. Un/inhibited organism: Political power, psychophysiology, and techniques.; 2009.

46. Wang W, Wen J, Luo ZG, Luo WY. How does environmental punishment affect regional green technology innovation?-Evidence from Chinese Provinces. Plos One. 2023;18(7). http://doi.org/10.1371/journal.pone.0288080

47. Liang LW, Wang ZB, Luo D, Wei Y, Sun JW. Synergy effects and it's influencing factors of China's high technological innovation and regional economy. Plos One. 2020;15(5). http://doi.org/10.1371/journal.pone.0231335

48. Yang HC, Li LS, Liu YB. The effect of manufacturing intelligence on green innovation performance in China. Technol Forecast Soc. 2022;178. http://doi.org/10.1016/j.techfore.2022.121569

49. Rehman SU, Ashfaq K, Bresciani S, Giacosa E, Mueller J. Nexus among intellectual capital, interorganizational learning, industrial Internet of things technology and innovation performance: a resource-based perspective. J Intellect Cap. 2023;24(2):509-34. http://doi.org/10.1108/JIC-03-2021-0095

50. Andersen SC, Jakobsen ML. Political Pressure, Conformity Pressure, and Performance Information as Drivers of Public Sector Innovation Adoption. Int Public Manag J. 2018;21(2):213-42. http://doi.org/10.1080/10967494.2018.1425227

51. He C, Wang YP, Tang K. Impact of Low-Carbon City Construction Policy on Green Innovation Performance in China. Emerg Mark Financ Tr. 2023;59(1):15-26. http://doi.org/10.1080/1540496X.2022.2089019

52. Xu JH, He SY. Can Grid Governance Fix the Party-state's Broken Windows? A Study of Stability Maintenance in Grassroots China. China Quart. 2022;251:843-65. http://doi.org/10.1017/S0305741022000509

53. Bianchi I. Empowering policies for grassroots welfare initiatives: Blending social innovation and commons theory. Eur Urban Reg Stud. 2023;30(2):107-20. http://doi.org/10.1177/09697764221129532

54. Qian X, Cai Y, Yin C. Driving Force of Grassroots Self-governance in Beijing's Neighborhoods: Social Capital, Community Network and Community Service Motivation. Lex Localis. 2019;17(1):159-77. http://doi.org/10.4335/17.1.159-177(2019)

4. Increased emphasis on research contributions/values, limitations and future research directions.

Answer: Thank you for your valuable advice, We have enriched the research contributions/values, limitations and future research directions. The specific changes and references are as follows:

This paper has the following shortcomings and provides directions for future research: firstly, this paper only considers preconditions such as technology, organisation and environment, but there are many factors that affect innovation performance. Future research can incorporate factors such as strategic orientation and organisational resources to study the influencing factors of innovation performance more comprehensively. Second, 35 grassroots social governance innovation cases were selected for this paper, and the generalisability of the results of the qualitative comparative analysis in terms of application is limited by the sample size. In the future, more data can be collected for further analysis. Therefore, this study has important theoretical and practical value, and can provide new ideas and methods for the research of management and social governance.

5. References must be further increased, especially those within the last three years.

Answer: Thanks for your suggestion, we have updated the content and references.

---

## [Decision Letter · Decision Letter 1]

15 Nov 2023

PONE-D-23-21194R1Research on the Improvement Path of Grassroots Social Governance Innovation Performance in China——Qualitative comparative analysis based on 35 casesPLOS ONE

Dear Dr. Chen,

Thank you for submitting your manuscript to PLOS ONE. After careful consideration, we feel that it has merit but does not fully meet PLOS ONE’s publication criteria as it currently stands. Therefore, we invite you to submit a revised version of the manuscript that addresses the points raised during the review process.

We look forward to receiving your revised manuscript.

Kind regards,

Chunyu Zhang

Academic Editor

PLOS ONE

**Additional Editor Comments:**

One reviewer suggested rejecting the manuscript.  According to PLoS one's publishing standards, if the reviewer's suggestions cannot be resolved, this manuscript cannot be accepted.  Of course, the novelty proposed by the reviewer can be ignored.

Reviewers' comments:

Reviewer's Responses to Questions

**Comments to the Author**

1. If the authors have adequately addressed your comments raised in a previous round of review and you feel that this manuscript is now acceptable for publication, you may indicate that here to bypass the “Comments to the Author” section, enter your conflict of interest statement in the “Confidential to Editor” section, and submit your "Accept" recommendation.

Reviewer #1: All comments have been addressed

Reviewer #3: (No Response)

Reviewer #4: All comments have been addressed

2. Is the manuscript technically sound, and do the data support the conclusions?

Reviewer #1: Yes

Reviewer #3: (No Response)

Reviewer #4: Yes

3. Has the statistical analysis been performed appropriately and rigorously? 

Reviewer #1: Yes

Reviewer #3: (No Response)

Reviewer #4: Yes

4. Have the authors made all data underlying the findings in their manuscript fully available?

Reviewer #1: Yes

Reviewer #3: (No Response)

Reviewer #4: Yes

5. Is the manuscript presented in an intelligible fashion and written in standard English?

Reviewer #1: Yes

Reviewer #3: (No Response)

Reviewer #4: Yes

6. Review Comments to the Author

Reviewer #1: The author of this paper has made careful revisions according to the review opinions, and has reached the publication level, and it is recommended to publish. Of course, the format or normative aspects still need to be modified according to the requirements of the editorial department.

Reviewer #3: The language of the paper needs to be more concise.

Regarding the data and methods, what is the data source? These aspects need more detailed explanation. There could also be graphical preseantations of the trend of variables.

The theoretical basis for the mechanisms presented in this study is very weak and unconvincing. The novelty of this manuscript is very inadequate. There are many similar studies. In addition, the research purpose and research framework of the manuscript are unclear. There are unaddressed issues in the paper that bear on the central conclusions of the paper.

Please follow the literature review by a clear and concise state of the art analysis. This should clearly show the knowledge gaps identified and link them to your paper goals. Please reason both the novelty and the relevance of your paper goals. Clearly discuss what the previous studies that you are referring to.What are the Research Gaps/Contributions? Please note that the paper may not be considered further without a clear research gap and novelty of the study. There is no flow in the text. It partly depends on the lack of proofreading but also on the fact that many statements and claims are made without being followed up by a clear and logical discussion. It is especially problematic in the Introduction that brings up a number of findings from different areas without linking them together.

Reviewer #4: (No Response)

7. PLOS authors have the option to publish the peer review history of their article (what does this mean?). If published, this will include your full peer review and any attached files.

Reviewer #1: **Yes: **liu bangfan

Reviewer #3: No

Reviewer #4: **Yes: **mohsin shahzad

---

## [Author Response · Author response to Decision Letter 1]

25 Dec 2023

Dear Editor,

I hope this message finds you well. I am writing to provide an update regarding the manuscript with the reference number PONE-D-23-21194R1, titled "Research on the Improvement Path of Grassroots Social Governance Innovation Performance in China - Qualitative comparative analysis based on 35 cases," which was submitted to PLOS ONE.

This article plays a key role in whether I can successfully graduate on time. I would like to express my gratitude for the careful consideration given to our manuscript. We have thoroughly reviewed the comments raised during the review process and have made significant revisions to address them.

I would like to bring to your attention that we recently received comments from Reviewer #3, which were not previously communicated to us. We apologize for any confusion caused by this oversight. However, I want to assure you that we have taken these comments into account and have made additional modifications to further strengthen the manuscript.

Thank you for your understanding and consideration. We are very fortunate if the revised manuscript now meets PLOS ONE's publication criteria. We look forward to receiving further guidance and instructions for the submission of the revised version.

Best wishes,

Nana Song，Longshun Xu，Xiansheng Chen，Huange Xu, Shuoliang Jiang

In response to Reviewer #3's concerns, we have made the following improvements:

1. The language of the paper needs to be more concise.

Answer: Thanks for your suggestion. We asked our English major friends to help us polish the whole article. Specific changes have been marked.

2.Regarding the data and methods, what is the data source? These aspects need more detailed explanation. There could also be graphical preseantations of the trend of variables.

Answer: We agree. An additional explanation is given in the paper：

“According to the above case selection criteria, this paper selects 35 grassroots social governance innovation cases from the "Top Ten Social Governance Innovation in China" award selected by the China All-round Well-off Forum and official news reports of governments at all levels. The data based on the values of each variable come from official data, journal data, and Internet media such as Xinhuanet, NetEase, Sohu and Ifeng.com”.

In addition, due to the word limit, the 35 cases selected were not shown in the paper, and the data provided clearly showed the source of the cases, as well as variable assignment and coding. Specific cases are as follows:

Table1: study case information

caseid Case

case1 Tong 'an District, Xiamen: Building a miniature "closed system" for community governance

case2 Rizhao City, Shandong Province: Build a four-level digital platform

case3 Wuchang District, Wuhan City, Hubei Province: "three-community cooperation" promotes diverse community governance

case4 Yangpu District, Shanghai: "Community Center"

case5 Huaian City, Jiangsu Province, Letters and Calls Bureau: "126" new mode of letters and calls work

case6 Xiangyang City, Hubei Province: public recommendation and competitive selection of county Party secretaries and county (district) chiefs

case7 Liupanshui City, Guizhou Province: The "Three iron measures" of political and legal organs serve the non-public economy to promote social harmony

case8 Zhejiang Provincial Civil Affairs Department: "Village rules promote harmony, social conventions ensure governance"

case9 The Organization Department of the CPC Shaoxing Municipal Committee: Party building leads the rural revitalization "five-star standard 3A striving" working mechanism

case10 Ningjiang District, Songyuan City, Jilin Province: Tuanjie Street strives to improve the community safety prevention system

case11 Harbin City, Heilongjiang Province: Community neighborly center practice innovation

case12 Deqing County, Zhejiang Province: urban and rural domestic waste treatment integration project

case13 Hangzhou, Zhejiang Province: Urban brain "Hangzhou Solution"

case14 Liyang City, Jiangsu Province: "people's hall" set up in every village

case15 Tianya District, Sanya City, Hainan Province: "Joint Command Center"

case16 Shenzhen Pingshan New District: the application of big data in social governance

case17 Xigang District of Dalian City: 365 work system

case18 Dongguan, Guangdong Province: Innovation and promotion of points-based management constantly improve the grass-roots governance system

case19 Jinan City, Shandong Province: Establish the "four social linkage" mechanism

case20 Nanjing, Jiangsu Province: Party Building Alliance of elderly care Service organizations

case21 Yinchuan City, Ningxia: Party building leads grassroots governance

case22 Inner Mongolia Autonomous Region: Building Safe Chifeng

case23 Chengdu, Sichuan Province: Tianfu citizen Cloud "Top ten" citizen reputation series service projects

case24 Shaoxing City, Zhejiang Province: remote convenience service center

case25 Wuxi City, Jiangsu Province: The first river chief system in China

case26 Wulong District of Chongqing: Optimizing grid governance to promote the construction of social governance system

case27 Jiuquan City, Gansu Province: implement the working mode of "one office, four rooms and one center"

case28 Donglan County, Hechi City, Guangxi Autonomous Region: Seven people, one station and one position working method

case29 Guizhou Sandu Zhonghe: Explore the "1+2" work method to build the "Five Bridges" to build the "five homes"

case30 Hefei, Anhui Province: Rule of law thinking improves people's livelihood

case31 Yan 'an City, Shaanxi Province: implement the "problem wall + echo wall" system

case32 Jinzhong City, Shanxi Province: "Five checks and five treatments" to improve the sense of gain in poverty alleviation

case33 Songgang Street, Baoan District, Shenzhen City, Guangdong Province: "One grid multiple" grid grassroots governance

case34 Jiangbei District, Ningbo City, Zhejiang Province: "One thing" integrated reform stimulates new momentum of grassroots governance

case35 Hunchun City, Yanbian Prefecture, Jilin Province: Build a team of professional community workers

The specific research method is further clearly pointed out in the paper, the qualitative comparative analysis method is used in this paper, and the advantages of the method, why the method is used, and the technical principle of the method are explained. The specific contents are as follows:

“In this paper, we abandon the traditional binary linear model and try to use fuzzy set qualitative comparative analysis (fsQCA) from the perspective of configuration to analyze the multiple driving path to improve the innovation performance of grassroots social governance. The qualitative comparative analysis method can solve the data analysis of 10-60 small and medium-sized samples, and find out the logical relationship between the innovation performance of grassroots social governance under different matching modes of ante-cause conditions. The fsQCA method has the following considerations”

3.The theoretical basis for the mechanisms presented in this study is very weak and unconvincing. The novelty of this manuscript is very inadequate. There are many similar studies.

Answer: Thanks for your suggestion. I have modified and supplemented the theoretical basis. Details are as follows:

“The academic circles mainly study the innovation of grassroots social governance from three perspectives。The first is to study the barriers to innovation in grassroots social governance from the perspective of problem interpretation.。From the government level, the government is the leader of grassroots social governance innovation, and grassroots social governance innovation has a strong political nature, and the government pays more attention to efficiency but ignores democracy[5].It makes the grassroots social governance innovation lack vitality. At the same time, the dislocation and absence of functions of grass-roots governments, insufficient supply of resources, and lagging management reduce the innovation performance to some extent[6].From the perspective of the public, grassroots social governance innovation is easy to be covered up by technocracy and exclusivity, and the public is rarely the core of previous deliberation, decision-making and development[7], and there is no institutional guarantee for public participation in grassroots social governance innovation; From the perspective of innovation itself, grassroots social governance innovation is not sustainable, innovation subjects lack enthusiasm, innovation projects lack operability, and the use of innovative technologies is insufficient. The second is to study the influencing factors of grassroots social governance innovation from the perspective of causality. Based on the political dimension, some scholars have discussed the impact of government authority, political support[8], organizational structure, power structure, legal structure, action structure, grassroots social governance system, and the "trinity" model on grassroots social governance innovation. Other scholars have studied the interactions between personal attributes, political leaders and their environment[9], top-down political mandates and policy pressures, civil rights and civic participation[10], and the overlap of innovative activities with existing social structures and social needs[11]. The net effect of other factors on grassroots social governance innovation. The third is to study the innovation path of grassroots social governance based on the goal-oriented perspective. From the government level, the government should optimize the grid management system in governance innovation, build the network structure of actors, and form innovation synergy[12].The government should further integrate various resources, respect the interests of multiple subjects, strengthen political commitment, optimize the collaborative governance model, improve the grassroots governance innovation system, and form a grassroots governance innovation community. From the public level, improve the enthusiasm of the public to participate in grassroots social governance innovation, improve the "government-society" cooperation mechanism, strengthen technological innovation and application, and stimulate the vitality of grassroots social governance innovation[13];From the perspective of the innovation process, the innovation process of grassroots social governance itself is conducive to fostering a more general democracy, which requires strengthening the use of technology, coordinating the interests of all parties, and improving infrastructure[14].The innovation process needs to be more open to public scrutiny and wider participation.”

We added the following literature:

“5. Stirling A. Towards Innovation Democracy? Participation, Responsibility and Precaution in Innovation Governance. Participation, Responsibility and Precaution in Innovation Governance.(November 2014). SWPS. 2014;24.

 6. Li A. Analysis of the Problems and Countermeasures of Grass-Roots Government in Community Governance. Academic Journal of Management and Social Sciences. 2023;5(1):131-7.

 7. Kearnes M, Chilvers J. Remaking Participation in Science and Democracy. Science, Technology, and Human Values. 2020;45(3).

 8. Cinar E, Trott P, Simms C. A systematic review of barriers to public sector innovation process. Public Manag Rev. 2019;21(2):264-90.

 9. Wynen J, Verhoest K, Ongaro E, Van Thiel S, In CWTC. Innovation-oriented culture in the public sector: Do managerial autonomy and result control lead to innovation? Public Manag Rev. 2014;16(1):45-66.

10. Crosby BC, T Hart P, Torfing J. Public value creation through collaborative innovation. Public Manag Rev. 2017;19(5):655-69.

11. Korac S, Saliterer I, Walker RM. Analysing the environmental antecedents of innovation adoption among politicians and public managers. Public Manag Rev. 2017;19(4):566-87.

12. Mittelstaedt JC. The grid management system in contemporary China: Grass-roots governance in social surveillance and service provision. China Inform. 2022;36(1):3-22.

13. Huang X, Zhou L. “Paired competition”: A new mechanism for the innovation of urban grassroots governance. Chinese Journal of Sociology. 2023;9(1):3-45.

14. Smith A, Stirling A. Innovation, sustainability and democracy: An analysis of grassroots contributions. Journal of Self-Governance and Management Economics. 2018;6(1):64-97.”

4.In addition, the research purpose and research framework of the manuscript are unclear. There are unaddressed issues in the paper that bear on the central conclusions of the paper.

Answer: Thank you for your valuable advice. We re-sorted out the research framework, and according to the research framework to conduct problem research, and finally analyze and solve the problem. In the introduction of this paper, based on the practice status of grassroots social governance innovation in China and the research status of grassroots social governance innovation theory, two research objectives of this paper are proposed. In addition, the purpose and framework of the research are further explained in the theoretical and analytical framework. Details are as follows:

“The first is to study the barriers to innovation in grassroots social governance from the perspective of problem interpretation.” “The second is to study the influencing factors of grassroots social governance innovation from the perspective of causality. Based on the political dimension, some scholars have discussed the impact of government authority, political support, organizational structure, power structure, legal structure, action structure, grassroots social governance system, and the "trinity" model on grassroots social governance innovation.”、“The third is to study the innovation path of grassroots social governance based on the goal-oriented perspective. From the government level, the government should optimize the grid management system in governance innovation, build the network structure of actors, and form innovation synergy”

Line 91-97 “Expanding existing studies from the three dimensions of technical factors, organizational factors and environmental factors, focusing on the single factor influence of grassroots social governance innovation performance, ignoring the lack of research on the combination of multiple factors, and then revealing the core influencing factors of grassroots social governance innovation to achieve high performance and the substitution effect between different paths. It provides reference for local governments to adopt differentiated innovation paths of grassroots social governance according to local conditions.”

The specific framework is as follows:

“2.1 Technical factors and grassroots social governance innovation performance

In order to achieve higher performance in grassroots social governance innovation, it is necessary to break through the problem of insufficient use of technology in the traditional governance model, which requires increasing technological innovation efforts and improving technological innovation ability. Especially with the rapid development of network information technology, governance technology has become an important tool to promote grassroots social governance innovation, and digital technology has become a key driving force to improve grassroots social governance innovation performance[20]. Based on the actual situation of China's grass-roots social governance, China's grass-roots population is large, grass-roots problems are complex, and it is more necessary to promote technological innovation and increase the application of technology, so as to improve the innovation performance of grass-roots social governance. Most scholars view input-output, cost-benefit and other perspectives[21]. Focus on whether technology application can reduce governance costs and whether technology empowerment can promote performance. The technology application of grassroots social governance needs to rely on the support of technological innovation ability and technical facilities. Generally speaking, technological innovation capability is the core competitiveness of regional economic development and grassroots governance, and it represents the breadth and depth of grassroots social governance innovation, while technological facility platform is an important form of technology application, and an important tool to facilitate grassroots social governance and improve grassroots social governance performance[22].

2.2 Organizational factors and grassroots social governance innovation performance

Grassroots social governance innovation behavior is the specific implementation of strategic decision-making, and grassroots social governance innovation strategic decision-making is decided by the government organization to a large extent, and will be promoted by the superior government, their own degree of attention and government levels. Under the Chinese political system, the willingness of grassroots governments to innovate and develop depends to a certain extent on the financial resources and policy support given by higher level governments. Top-down political pressure is an important driver of innovation in public sector governance[23]. Therefore, grassroots social governance innovation cannot be achieved without the support of the higher level government, and the capital investment, policy inclination and resource allocation of the higher level government are the effective support for achieving grassroots social governance innovation[24]. Local governments' self-importance is the internal driving force for promoting innovation in grassroots social governance. Existing studies start with institutional analysis and development framework, and depict the relationship between the government's self-importance and innovation performance in terms of rule supply, policy formulation, policy implementation, and policy supervision[25]. In China's bureaucratic administrative management system, lower-level governments are controlled by higher-level governments, and governments at different levels have different control effectiveness over authoritative resources. Therefore, government levels largely determine the success or failure and effectiveness of grassroots social governance innovation.

2.3 Environmental factors and grassroots social governance innovation performance

Grassroots social governance innovation is inevitably influenced by environmental factors. Since organizations are embedded in the social environment, their basic goal is to adapt, survive and develop their institutional environment to adapt to the changing environment[26]. Based on institutionalism, peer and superior pressure, public pressure, and competitive pressure significantly affect governance performance and innovation level[27]. Some studies have shown that grassroots social governance innovation will be affected by the pressure of inter-governmental competition. In innovation activities, the higher the innovation level of neighboring governments, the greater the pressure on them to promote their own innovation practices, which will lead to the strengthening of innovation practices and improvement of innovation level of governments in the region[28]. In China, local governments respond to public demands in accordance with the requirements of the central government under the pressure of assessment, and the public puts pressure on the government based on efficiently solving the problems of grassroots governance and safeguarding their legitimate rights and interests. The pressure of public demand urges government departments to adopt innovative strategies to improve the performance of grassroots social governance[29]. Open government can be seen as a form of innovation in the public sector, as it requires the government to adopt new technologies with an open attitude[30]. Therefore, open government is an important guarantee for promoting innovation in grassroots social governance.”

We added the following literature:

“20. Sachan A, Kumar R, Kumar R. Examining the impact of e-government service process on user satisfaction. Journal of Global Operations and Strategic Sourcing. 2018;11(3):321-36.

21. Lee S, Jung K. The Role of community-led governance in innovation diffusion: The case of RFID Waste pricing system in the Republic of Korea. Sustainability-Basel. 2018;10(9):3125.

22. Saxena S. Summarizing the decadal literature in open government data (OGD) research: a systematic review. foresight. 2018;20(6):648-64.

23. Andersen SC, Jakobsen ML. Political Pressure, Conformity Pressure, and Performance Information as Drivers of Public Sector Innovation Adoption. Int Public Manag J. 2018;21(2):213-42. http://doi.org/10.1080/10967494.2018.1425227

24. Donegan M, Lester TW, Lowe N. Striking a balance: A national assessment of economic development incentives. Urban Aff Rev. 2021;57(3):794-819.

25. Jin P, Mangla SK, Song M. The power of innovation diffusion: How patent transfer affects urban innovation quality. J Bus Res. 2022;145:414-25.

26. Adnan HR, Hidayanto AN, Kurnia S. Citizens’ or government’s will? Exploration of why indonesia’s local governments adopt technologies for open government. Sustainability-Basel. 2021;13(20):11197.

27. Jing Y. Study on the influencing factors of e-government service capacity of provincial governments in China. Front. Edu. Res. 2020;3:12-5.

28. Manyika J, Lund S, Bughin J. Digital globalization: The new era global flows.: McKinsey Global Institute; 2016.

29. Cinar E, Trott P, Simms C. A systematic review of barriers to public sector innovation process. Public Manag Rev. 2019;21(2):264-90.

30. Myeong S, Ahn MJ, Kim Y, Chu S, Suh W. Government Data Performance: The Roles of Technology, Government Capacity, and Globalization through the Effects of National Innovativeness. Sustainability-Basel. 2021;13(22):12589.”

5. Please follow the literature review by a clear and concise state of the art analysis. This should clearly show the knowledge gaps identified and link them to your paper goals. Please reason both the novelty and the relevance of your paper goals. 

Answer: Thanks for your suggestion.

 First, in the research background of the first paragraph in the introduction, this paper makes a clear and concise analysis of the current situation of grassroots social governance in China, and puts forward the first research goal.

 China has made significant efforts to promote grassroots social governance innovation in order to modernize its social governance system and enhance its governance capacity. This has been further advanced by the establishment of awards such as the "China Local Government Innovation Award" and the "China Top Ten Social Governance Innovation Awards". Although governments at all levels have promoted the practice of grassroots social governance innovation, due to the imperfect system and mechanism, imperfect laws and regulations, and blocked participation channels, some grassroots social governance innovation projects have a short-lived phenomenon, and it is difficult to ensure the efficiency and sustainability of innovation projects[4]. Therefore, the first goal that this article addresses is: why do grassroots social governance innovation projects suffer from the same fate of being ineffective despite taking different paths? How to improve the practical efficiency of grassroots social governance innovation projects?

Second，After literature review, this paper studies the current situation of the relevant theories of grassroots social governance innovation performance, and puts forward the second research goal.

Scholars have explored the innovation of grassroots social governance from different perspectives, which is of reference significance to this paper. But there are two limitations to the existing research: First, most researchers only study the innovation performance of grassroots social governance from a single factor, and cannot clearly identify the "interactive relationship" between different factors. This situation often leads to excessive dependence on a single factor in the innovation performance of grassroots social governance, and it is impossible to maximize the efficiency of resource utilization through reasonable allocation of resources. Second, few studies have explored the joint effects of multiple factors, nor have in-depth studies been conducted on the anthems of the differences in grassroots social governance innovation performance, the lack of analysis on the interaction mechanism of multiple complex conditions from the perspective of configuration, the configuration law of conditions is not clear, and the differentiation path and combination of improving grassroots social governance innovation performance need to be studied. Therefore, the second goal that this article addresses is whether and to what extent multiple factors are necessary conditions for achieving high-performance grassroots social governance innovation. How do multiple factors couple and interact to enhance grassroots social governance innovation performance? 

6. Clearly discuss what the previous studies that you are referring to. What are the Research Gaps/Contributions? Please note that the paper may not be considered further without a clear research gap and novelty of the study.

Answer: Thanks for your suggestion. We have reviewed the literature and linked the literature with our research to highlight the value contribution and novelty of this research. The specific contents are as follows:

“The academic circles mainly study the innovation of grassroots social governance from three perspectives. The first is to study the barriers to innovation in grassroots social governance from the perspective of problem interpretation. From the government level, the government is the leader of grassroots social governance innovation, and grassroots social governance innovation has a strong political nature, and the government pays more attention to efficiency but ignores democracy[5].It makes the grassroots social governance innovation lack vitality. At the same time, the dislocation and absence of functions of grass-roots governments, insufficient supply of resources, and lagging management reduce the innovation performance to some extent[6].From the perspective of the public, grassroots social governance innovation is easy to be covered up by technocracy and exclusivity, and the public is rarely the core of previous deliberation, decision-making and development[7], and there is no institutional guarantee for public participation in grassroots social governance innovation; From the perspective of innovation itself, grassroots social governance innovation is not sustainable, innovation subjects lack enthusiasm, innovation projects lack operability, and the use of innovative technologies is insufficient. The second is to study the influencing factors of grassroots social governance innovation from the perspective of causality. Based on the political dimension, some scholars have discussed the impact of government authority, political support[8], organizational structure, power structure, legal structure, action structure, grassroots social governance system, and the "trinity" model on grassroots social governance innovation. Other scholars have studied the interactions between personal attributes, political leaders and their environment[9], top-down political mandates and policy pressures, civil rights and civic participation[10], and the overlap of innovative activities with existing social structures and social needs[11]. The net effect of other factors on grassroots social governance innovation. The third is to study the innovation path of grassroots social governance based on the goal-oriented perspective. From the government level, the government should optimize the grid management system in governance innovation, build the network structure of actors, and form innovation synergy[12].The government should further integrate various resources, respect the interests of multiple subjects, strengthen political commitment, optimize the collaborative governance model, improve the grassroots governance innovation system, and form a grassroots governance innovation community. From the public level, improve the enthusiasm of the public to participate in grassroots social governance innovation, improve the "government-society" cooperation mechanism, strengthen technological innovation and application, and stimulate the vitality of grassroots social governance innovation[13];From the perspective of the innovation process, the innovation process of grassroots social governance itself is conducive to fostering a more general democracy, which requires strengthening the use of technology, coordinating the interests of all parties, and improving infrastructure[14].The innovation process needs to be more open to public scrutiny and wider participation.”

We added the following literature:

5. Stirling A. Towards Innovation Democracy? Participation, Responsibility and Precaution in Innovation Governance. Participation, Responsibility and Precaution in Innovation Governance.(November 2014). SWPS. 2014;24.

 6. Li A. Analysis of the Problems and Countermeasures of Grass-Roots Government in Community Governance. Academic Journal of Management and Social Sciences. 2023;5(1):131-7.

 7. Kearnes M, Chilvers J. Remaking Participation in Science and Democracy. Science, Technology, and Human Values. 2020;45(3).

 8. Cinar E, Trott P, Simms C. A systematic review of barriers to public sector innovation process. Public Manag Rev. 2019;21(2):264-90.

 9. Wynen J, Verhoest K, Ongaro E, Van Thiel S, In CWTC. Innovation-oriented culture in the public sector: Do managerial autonomy and result control lead to innovation? Public Manag Rev. 2014;16(1):45-66.

10. Crosby BC, T Hart P, Torfing J. Public value creation through collaborative innovation. Public Manag Rev. 2017;19(5):655-69.

11. Korac S, Saliterer I, Walker RM. Analysing the environmental antecedents of innovation adoption among politicians and public managers. Public Manag Rev. 2017;19(4):566-87.

12. Mittelstaedt JC. The grid management system in contemporary China: Grass-roots governance in social surveillance and service provision. China Inform. 2022;36(1):3-22.

13. Huang X, Zhou L. “Paired competition”: A new mechanism for the innovation of urban grassroots governance. Chinese Journal of Sociology. 2023;9(1):3-45.

14. Smith A, Stirling A. Innovation, sustainability and democracy: An analysis of grassroots contributions. Journal of Self-Governance and Management Economics. 2018;6(1):64-97.

Explanatory note：

Based on this, the main contributions of this paper are as follows: First of all, from the perspective of configuration, this paper discusses the impact of coordination and integration of multiple factors on the innovation performance of grassroots social governance, expands the study based on single variable "net effect" to the study of multi-factor comprehensive effect based on the perspective of configuration, and clarifies the impact of each condition on the innovation performance of grassroots social governance and the linkage interaction between conditions. Secondly, the TOE framework is introduced in this paper to explore the influencing factors of grassroots social governance innovation performance from three aspects: technology, organization and environment, which is conducive to building a multidimensional comprehensive governance model of grassroots social governance innovation. Finally, through configuration analysis, this paper explores the anthegen configuration of the high innovation performance path of grassroots social governance, reveals its core influencing factors and the substitution effect between different paths, proves that there is not only one mode to improve grassroots social governance innovation performance, and provides new ideas for managers to flexibly choose innovation modes and improve grassroots social governance innovation performance.

6. There is no flow in the text. It partly depends on the lack of proofreading but also on the fact that many statements and claims are made without being followed up by a clear and logical discussion. It is especially problematic in the Introduction that brings up a number of findings from different areas without linking them together.

Answer: Thanks for your suggestion. In this paper, the introduction is reorganized and written. Some statements and claims are clearly discussed and linked together. And other parts of the content to sort out and clear ideas. Details are as follows: 

“The academic circles mainly study the innovation of grassroots social governance from three perspectives.

The first is to study the barriers to innovation in grassroots social governance from the perspective of problem interpretation. From the government level, the government is the leader of grassroots social governance innovation, and grassroots social governance innovation has a strong political nature, and the government pays more attention to efficiency but ignores democracy[5].It makes the grassroots social governance innovation lack vitality. At the same time, the dislocation and absence of functions of grass-roots governments, insufficient supply of resources, and lagging management reduce the innovation performance to some extent[6].From the perspective of the public, grassroots social governance innovation is easy to be covered up by technocracy and exclusivity, and the public is rarely the core of previous deliberation, decision-making and development[7], and there is no institutional guarantee for public participation in grassroots social governance innovation; From the perspective of innovation itself, grassroots social governance innovation is not sustainable, innovation subjects lack enthusiasm, innovation projects lack operability, and the use of innovative technologies is insufficient. The second is to study the influencing factors of grassroots social governance innovation from the perspective of causality. Based on the political dimension, some scholars have discussed the impact of government authority, political support[8], organizational structure, power structure, legal structure, action structure, grassroots social governance system, and the "trinity" model on grassroots social governance innovation. Other scholars have studied the interactions between personal attributes, political leaders and their environment[9], top-down political mandates and policy pressures, civil rights and civic participation[10], and the overlap of innovative activities with existing social structures and social needs[11]. The net effect of other factors on grassroots social governance innovation. The third is to study the innovation path of grassroots social governance based on the goal-oriented perspective. From the government level, the government should optimize the grid management system in governance innovation, build the network structure of actors, and form innovation synergy[12].The government should further integrate various resources, respect the interests of multiple subjects, strengthen political commitment, optimize the collaborative governance model, improve the grassroots governance innovation system, and form a grassroots governance innovation community. From the public level, improve the enthusiasm of the public to participate in grassroots social governance innovation, improve the "government-society" cooperation mechanism, strengthen technological innovation and application, and stimulate the vitality of grassroots social governance innovation[13];From the perspective of the innovation process, the innovation process of grassroots social governance itself is conducive to fostering a more general democracy, which requires strengthening the use of technology, coordinating the interests of all parties, and improving infrastructure[14].The innovation process needs to be more open to public scrutiny and wider participation.”

We added the following literature:

5. Stirling A. Towards Innovation Democracy? Participation, Responsibility and Precaution in Innovation Governance. Participation, Responsibility and Precaution in Innovation Governance.(November 2014). SWPS. 2014;24.

 6. Li A. Analysis of the Problems and Countermeasures of Grass-Roots Government in Community Governance. Academic Journal of Management and Social Sciences. 2023;5(1):131-7.

 7. Kearnes M, Chilvers J. Remaking Participation in Science and Democracy. Science, Technology, and Human Values. 2020;45(3).

 8. Cinar E, Trott P, Simms C. A systematic review of barriers to public sector innovation process. Public Manag Rev. 2019;21(2):264-90.

 9. Wynen J, Verhoest K, Ongaro E, Van Thiel S, In CWTC. Innovation-oriented culture in the public sector: Do managerial autonomy and result control lead to innovation? Public Manag Rev. 2014;16(1):45-66.

10. Crosby BC, T Hart P, Torfing J. Public value creation through collaborative innovation. Public Manag Rev. 2017;19(5):655-69.

11. Korac S, Saliterer I, Walker RM. Analysing the environmental antecedents of innovation adoption among politicians and public managers. Public Manag Rev. 2017;19(4):566-87.

12. Mittelstaedt JC. The grid management system in contemporary China: Grass-roots governance in social surveillance and service provision. China Inform. 2022;36(1):3-22.

13. Huang X, Zhou L. “Paired competition”: A new mechanism for the innovation of urban grassroots governance. Chinese Journal of Sociology. 2023;9(1):3-45.

14. Smith A, Stirling A. Innovation, sustainability and democracy: An analysis of grassroots contributions. Journal of Self-Governance and Management Economics. 2018;6(1):64-97.

---

## [Editor Report · Decision Letter 2]

9 Jan 2024

Research on the Improvement Path of Grassroots Social Governance Innovation Performance in China——Qualitative comparative analysis based on 35 cases

PONE-D-23-21194R2

Dear Dr. Chen,

We’re pleased to inform you that your manuscript has been judged scientifically suitable for publication and will be formally accepted for publication once it meets all outstanding technical requirements.

Kind regards,

Chunyu Zhang

Academic Editor

PLOS ONE
---

## [Editor Report · Acceptance letter]

29 Jan 2024

PONE-D-23-21194R2 

PLOS ONE

Dear Dr. Chen, 

I'm pleased to inform you that your manuscript has been deemed suitable for publication in PLOS ONE. Congratulations! Your manuscript is now being handed over to our production team.

Kind regards, 

on behalf of

Dr. Chunyu Zhang 

Academic Editor

PLOS ONE